# Exploring Ozone-climate Interactions in Idealized CMIP6 DECK Experiments

Jingyu Wang[1,2], Gabriel Chiodo[1,3], Timofei Sukhodolov[4], Blanca Ayarzagüena[5], William T. Ball[†], Mohamadou Diallo[6], Birgit Hassler[7], James Keeble[8], Peer Nowack[9], Clara Orbe[10], and Sandro Vattioni[1]

[1]Institute for Atmospheric and Climate Science, ETH Zurich, Zurich, Switzerland
[2]Lunar & Planetary Laboratory/Department of Planetary Sciences, University of Arizona, Tucson AZ, USA
[3]Instituto de Geociencias, CSIC (IGEO-CSIC), Madrid, Spain
[4]Physikalisch-Meteorologisches Observatorium Davos and World Radiation Center, Davos, Switzerland
[5]Facultad de CC Fisicas, Universidad Complutense de Madrid, Madrid, Spain
[6]Forschungszentrum Juelich, Germany
[7]Deutsches Zentrum für Luft- und Raumfahrt, Institut für Physik der Atmosphäre, Oberpfaffenhofen, Germany
[8]Lancaster Environment Center, Lancaster University, Lancaster, UK
[9]Institute of Theoretical Informatics & Institute of Meteorology and Climate Research (IMK-ASF), Karlsruhe Institute of Technology, Germany
[10]NASA Goddard Institute for Space Studies, New York, USA
[†]deceased

**Correspondence:** Jingyu Wang (wangjingyu@arizona.edu), Gabriel Chiodo (gabriel.chiodo@csic.es)

## Abstract

Under climate change driven by increased carbon dioxide ($CO_2$) concentrations, stratospheric ozone will respond to temperature and circulation changes, leading to chemistry-climate feedback by modulating large-scale atmospheric circulation and Earth's energy budget. However, there is a significant model uncertainty since many processes are involved and few models have a detailed chemistry scheme. This work employs the latest data from Coupled Model Intercomparison Project Phase 6 (CMIP6) to investigate the ozone response to increased $CO_2$. We find that in most models, ozone increases in the upper stratosphere (US) and extratropical lower stratosphere (LS), and decreases in the tropical LS, thus the total column ozone (TCO) response is small in the tropics. The ozone response is mainly driven by the slower chemical destruction cycles in the US and enhanced upwelling in the LS, with a highly model-dependent Arctic ozone response to polar vortex strength changes. We then explore the ozone-climate feedback, by combining offline calculations and comparisons between models with ("chem") and without ("no-chem") interactive chemistry. We find that the stratospheric temperature response is substantial, with a global negative radiative forcing ranging from -0.03 W m$^{-2}$ to -0.19 W m$^{-2}$. We find that chem models consistently simulate less tropospheric warming and strong weakening of the polar stratospheric vortex, which results in a larger increase of sudden stratospheric warming (SSW) frequency than in most no-chem models. Our findings show that ozone-climate feedback is essential for the climate system and should be considered in the development of Earth System Models.

## 1 Introduction

Stratospheric ozone abundances are sensitive to temperature and circulation variations and, thus, respond to climate change (Shepherd, 2008). In turn, ozone variations can also affect temperature and the large-scale atmospheric circulation, as ozone is a radiatively active gas that absorbs solar and terrestrial radiation, and radiatively heats the stratosphere (Brasseur and Solomon, 2005). Therefore, under externally-forced climate change, stratospheric ozone will respond to and in turn feedback onto climate. Understanding the mechanisms driving the ozone response and its implications for climate, and in particular the uncertainty across models in this feedback, is critical for future climate projections.

The distribution of ozone is primarily determined by production and loss from chemical reactions and transport processes (Brasseur and Solomon, 2005). Chemistry-climate models are often employed to simulate the ozone distribution and analyze the response of ozone and its feedback on climate. From a chemical perspective, two major factors influence the changes in production and destruction of ozone due to changes in atmospheric abundances of greenhouse gases (GHGs). Firstly, an increase in the abundance of GHGs, most importantly $CO_2$, leads to cooling of the stratosphere. In particular in the upper stratosphere (US) where the role of transport is comparatively less important due to the short chemical lifetime of ozone, this cooling slows down the destruction of ozone due to the positive temperature dependence of the catalytic cycles (Barnett et al., 1975; Haigh and Pyle, 1982) and also reduces odd oxygen loss. Therefore, local (radiative) cooling leads to a net increase in ozone mixing ratios. Secondly, since the loss rate of odd oxygen is proportional to the abundance of atomic oxygen, the increased efficiency of the three-body reaction $O_2 + O + M \rightarrow O_3 + M$ reduces the atomic oxygen abundance and consequently reduces odd oxygen loss (Jucks and Salawitch, 2000; Jonsson et al., 2004). Conversely, in the lower stratosphere (LS), the larger ozone column abundance overhead acts as a shield, reducing the sunlight responsible for ozone production. As a result of this self-shielding effect, the abundance of ozone decreases in this region (Haigh and Pyle, 1982; Jonsson et al., 2004; Meul et al., 2014; Keeble et al., 2017).

From a transport perspective, the expansion of the tropopause due to warming at the surface will effectively replace ozone-rich stratospheric air with ozone-poor tropospheric air, leading to a decrease of ozone near the tropopause. The warming in the tropical troposphere also accelerates the subtropical jets and results in a faster Brewer-Dobson Circulation (BDC), which means faster tropical upwelling and poleward transport, resulting in more efficient transport of ozone from the tropics to the extratropics in the LS (Butchart, 2014; Abalos et al., 2021). This results in a change in the latitudinal distribution of ozone. Such transport effects on ozone are more pronounced in the lower/middle stratosphere, while chemical effects dominate in its upper part (Oman et al., 2010; Zubov et al., 2012). Because of the different sign of the ozone change in the US and LS, the total column ozone (TCO) response in the tropics depends on the opposing impacts from different dominant mechanisms. When modeled, the representation and relative importance of these processes can be model dependent, resulting in different sign of the low-latitude TCO response (Oman et al., 2010; Chiodo et al., 2018; Keeble et al., 2021). The stratospheric polar vortex is also an important feature that shapes the distribution of ozone, since it creates a transport barrier between mid-latitudes and the

poles of the winter hemisphere (Shepherd, 2008; Seinfeld et al., 1998). The response of the polar vortex to increased $CO_2$ level is very uncertain, with a large divergence in the projection of polar vortex strength in winter across climate model ensembles, which will affect the mixing of ozone-poor polar air with ozone-rich air in lower latitudes (Ayarzagüena et al., 2020; Karpechko et al., 2022).

Many factors can influence the ozone response to climate change, making it difficult to separate the influence of individual forcing agents. First, because of their role as sources of radical species in the stratosphere, the different concentrations of $CH_4$ and $N_2O$ used in inter-model comparisons may potentially offset the effects of $CO_2$ (Revell et al., 2012). Second, comparing different future scenarios may be non-trivial due to the non-linearity from the combined effects of ozone-depleting substances (ODSs), GHGs, and ozone precursors (Meul et al., 2015). $CO_2$ is the only forcing considered for diagnosing transient and equilibrium climate model sensitivity (TCR and ECS), and, thus, a large amount of data is available for this individual forcing in the past multi-model comparison.

Previous studies comparing simulations of a small set of models found that under increasing $CO_2$, ozone mixing ratios will increase in the US, decrease in the tropical LS and increase in the extratropical LS (Oman et al., 2010; Chiodo et al., 2018). Aside from documenting changes in ozone, previous research has also highlighted another crucial element in the ozone-climate interaction: changes in ozone can potentially lead to a chemistry-climate feedback, with implications for the modeled changes in the tropospheric and surface climate. It has been suggested that the ozone-climate feedback may affect both climate sensitivity (Dietmüller et al., 2014; Muthers et al., 2014, 2015; Nowack et al., 2018; Hardiman et al., 2019) and dynamical sensitivity (Chiodo and Polvani, 2017; Nowack et al., 2017, 2018; Chiodo and Polvani, 2019; Orbe et al., 2024). In addition, it may play a role in modulating the modeled response of El Niño–Southern Oscillation (ENSO) to global warming (Nowack et al., 2017) and in dampening the climate system response to solar forcing (Chiodo and Polvani, 2016; Muthers et al., 2016). Furthermore, the importance of chemistry feedback has been shown for the stratosphere-troposphere coupling (Haase and Matthes, 2019), in particular for the Arctic climate (Friedel et al., 2022) and the Southern annular mode (Morgenstern, 2021). Consequently, implementing interactive chemistry, i.e. an online scheme (either a full chemical submodel or a linearized ozone scheme) that allows to capture the feedbacks between chemistry and temperature/dynamics, in Earth System Models (ESM) is thought to be important for climate projections.

However, there remains a large inter-model discrepancy in both the ozone response and its climate feedback, and reasons for this uncertainty are still unclear. For instance, the magnitude and peak location of stratospheric ozone response have a notable inter-model discrepancy, which can also lead to a spread in stratospheric cooling (Chiodo et al., 2018). Simulated tropical TCO shows a significant inter-model spread in its magnitude and sign of the response, which stems mainly from the tropical lower stratospheric ozone (LSO3). The one possible source of the inter-model discrepancy in the tropics is the spread of the strengthening of the ascending branch of the BDC (Chiodo et al., 2018). The magnitude of the ozone-climate feedback, quantified as the impact of interactive ozone chemistry on the global-mean surface air temperature response to abrupt quadrupling of $CO_2$, ranges from 20% (Nowack et al., 2015), to 7%-8% (Dietmüller et al., 2014; Muthers et al., 2016), to nil (Marsh et al., 2016; Chiodo and Polvani, 2019). However, these previous studies have relied either on individual model simulations or a maximum of three models.

Considering the limited number of models analyzed in the past, it is necessary to compare the ozone response and associated climate feedback across more models and additional idealized scenarios (including transient experiments). This is now possible given that the number of models employing interactive chemistry schemes has substantially increased (by factor of three) in the last Coupled Model Intercomparison Project Phase 6 (CMIP6) (Keeble et al., 2021) compared to the previous phase 5 (Chiodo et al., 2018).

This work analyzes the latest CMIP6 dataset, which has more and updated models and a larger range of ECS compared to CMIP5 (Flynn and Mauritsen, 2020). Specifically, three 150 year-long Diagnostic, Evaluation and Characterization of Klima (DECK) experiments which are piControl, abrupt-4×$CO_2$ and 1pct$CO_2$ (increase up to 4×$CO_2$), are used to analyze the response and potential drivers of ozone response to elevated $CO_2$ (Eyring et al., 2016). The linearity of ozone response to $CO_2$ increase can be investigated by comparing results from the two increased $CO_2$ scenarios. We also use the data from the same three 150 year-long time-slice experiments from the new version of our in-house model, SOlar Climate Ozone Links v4.0 (SOCOLv4), under piControl (1850) CMIP6 boundary conditions. The climate impact is then investigated by conducting offline calculations of radiative transfer using Parallel Offline Radiative Transfer (PORT) for one year and prescribing the ozone mixing ratio the same as that from CMIP6 abrupt-4×$CO_2$ experiments to get the radiative perturbation. Lastly, by grouping models based on whether interactive chemistry is employed, comparison between chem and no-chem models is conducted through the investigation of the response in temperature and circulation between these two categories.

The structure of this paper is as follows: Section 2 introduces the data and models used in this work, Section 3 presents the main findings, and Section 4 concludes and discusses the findings and broader implications.

## 2 Data and Models

### 2.1 Data

Data analyzed in this research are from the DECK experiments of CMIP6, including piControl, abrupt-4×CO$_2$ and 1pctCO$_2$ experiments. There are in total 20 models which have ozone data for piControl, 20 models for abrupt-4×CO$_2$ and 19 models for 1pctCO$_2$. Among the 20 models, there are 13 chem models that performed abrupt-4×CO$_2$ and 12 chem models which performed 1pctCO$_2$ (see Table 1). We only use 7 no-chem models that have chem counterpart (see below). Data within the time-period of 150 years from the start of each experiment are extracted. The number of ensemble members in each experiment of each model varies, thus we only use one ensemble member with the same physics version ("p") for the analysis within each model. Note that zonal wind and vertical velocity data from MRI-ESM2-0 are not included because of the large departure from the data of other models with no reasonable trend. Furthermore, data of the first 15 years of ozone mixing ratio from the GISS-E2-1-G piControl experiment are not used due to an existing trend in this period, indicating that the model hadn't finished the spin-up phase. Brief descriptions of CMIP6 models are proved in Keeble et al. (2021) with exceptions of SOCOL and GISS-E2-1-G, which we provide below.

SOCOLv4 (Sukhodolov et al., 2021) is based on the combination of the MPIMET (Hamburg, Germany) Earth System model (MPI-ESM1-2LR, Mauritsen et al. (2019)) consisting of ECHAM6 for atmosphere and MPIOM (Jungclaus et al., 2013) for the ocean as well as JSBACH for terrestrial biosphere and HAMOCC for the ocean's biogeochemistry with the latest versions of the chemical (MEZON) (Egorova et al., 2003) and microphysical (AER) (Sheng et al., 2015) modules. SOCOLv4 uses the low-resolution (LR) configuration of the MPI-ESM model, which corresponds to a spectral truncation at T63, providing an approximate horizontal grid spacing of 1.9°×1.9°. The vertical resolution of the atmosphere is set to 47 levels from the surface to 0.01 hPa, using a hybrid sigma-pressure coordinate system. Besides SOCOLv4, we also use the results form its earlier CMIP5 version SOCOL-MPIOM (Muthers et al., 2014) to compare the ozone response within the SOCOL model family. Boundary conditions for SOCOLv4 simulations follow the recommendations of Eyring et al. (2016), with the exception being the volcanic forcing, which is set to the quiet conditions of year 2000 instead of the 1850-2014 mean. Initial conditions for the coupled atmosphere-ocean system have been taken from the piControl simulation of MPI-ESM1-2-LR.

GISS-E2-1 (Kelley et al., 2020) consists of an atmosphere component and an ocean component. The horizontal and vertical resolution of the atmospheric component is 2°latitude by 2.5°longitude with 40 vertical layers from the surface to 0.1 hPa in the lower mesosphere. Among the two versions used in this work, atmospheric composition is prescribed in the first version, denoted in the CMIP6 archive as physics-version=1 ("p1"). In the other version ("p3"), ozone is calculated prognostically using the One-Moment Aerosol (OMA) model. Each of the two physics versions of the GISS-E2-1 atmospheric component is coupled to the ocean general circulation models GISS Ocean version 1 (GO1). It has a horizontal resolution of 1°latitude by 1.25°longitude and 40 vertical layers. We also examine results from the high vertical resolution version of the GISS CMIP6 climate model submission, GISS-E2-2 (Rind et al., 2020; Orbe et al., 2020). Though identical in horizontal resolution to E2-1, E2-2 has more than twice the number of vertical levels (102) and a higher model top (0.002 hPa). This, in combination a non-

**Table 1.** Summary of the data from CMIP6 and SOCOL experiments used in this research (Keeble et al., 2021). For models without interactive chemistry, ozone fields in most models are prescribed using CMIP6 dataset, with the exception of CESM2 and CESM-FV2, which use simulations performed with the CESM2-WACCM model; for GISS-E2-1-G (p1) and GISS-E2-2-G (p1), it is prescribed with the offline ozone fields from GISS-E2-1-G (p3) and GISS-E2-2-G (p3), respectively (Kelley et al., 2020); for HadGEM3-GC31-LL in increased $CO_2$ experiments, the ozone field is vertically redistributed from the piControl field to account for the tropopause shift (Hardiman et al., 2019).

| Model | Horizontal Resolution (lon×lat) | Vertical Resolution | Interactive Chemistry (Y/N) | Ozone Scheme | piControl/ abrupt-4×CO₂/ 1pctCO₂ (year) |
|---|---|---|---|---|---|
| CESM2 | 288×192 | 32 levels; top level 2.25 hPa | N | Prescribed | 150/150/150 |
| CESM2-FV2 | 144×96 | 32 levels; top level 2.25 hPa | N | Prescribed | 150/150/150 |
| CESM2-WACCM | 288×192 | 70 levels; top level $4.5 \times 10^{-6}$ hPa | Y | Interactive chemistry | 150/150/150 |
| CESM2-WACCM-FV2 | 144×96 | 70 levels; top level $4.5 \times 10^{-6}$ hPa | Y | Interactive chemistry | 150/150/150 |
| CNRM-CM6-1 | 256×128 | 91 levels; top level 78.4 km | Y | Simplified online scheme | 150/150/150 |
| CNRM-CM6-1-HR | 720×360 | 91 levels; top level 78.4 km | Y | Simplified online scheme | 150/150/150 |
| CNRM-ESM2-1 | 256×128 | 91 levels; top level 78.4 km | Y | Interactive chemistry | 150/150/150 |
| E3SM-1-0 | 360×180 | 72 levels; top level 0.1 hPa | Y | Simplified online scheme | 150/150/150 |
| GFDL-CM4 | 360×180 | 33 levels; top level 1 hPa | N | Prescribed (CMIP6 dataset) | 150/150/150 |
| GFDL-ESM4 | 288×180 | 49 levels; top level 1 Pa | Y | Interactive chemistry | 150/150/150 |
| GISS-E2-1-G (p1) | 144×90 | 40 levels; top level 0.1 hPa | N | Prescribed | 150/150/150 |
| GISS-E2-1-G (p3) | 144×90 | 40 levels; top level 0.1 hPa | Y | Interactive chemistry | 150/150/150 |
| GISS-E2-2-G (p1) | 144×90 | 102 levels; top level 0.002 hPa | N | Prescribed | 150/150/150 |
| GISS-E2-2-G (p3) | 144×90 | 102 levels; top level 0.002 hPa | Y | Interactive chemistry | 150/150/150 |
| HadGEM3-GC31-LL | 192×144 | 85 levels; top level 85km | N | Prescribed | 150/150/150 |
| MPI-ESM1-2-LR | 192×96 | 47 levels; top level 0.01 hPa | N | Prescribed (CMIP6 dataset) | 150/150/150 |
| MRI-ESM2-0 | 128×64 | 80 levels; top level 0.01 hPa | Y | Interactive chemistry | 150/150/150 |
| SOCOL-MPIOM | 96×48 | 47 levels; top level 0.01 hPa | Y | Interactive chemistry | 150/150/– |
| SOCOLv4 | 192×96 | 47 levels; top level 0.01 hPa | Y | Interactive chemistry | 150/150/150 |
| UKESM1-0-LL | 192×144 | 85 levels; top level 85km | Y | Interactive chemistry | 150/150/150 |

orographic gravity wave drag scheme that is directly tethered to parameterized convection, produces in E2-2 more credible middle atmosphere dynamical and transport circulations, compared to observations (Orbe et al., 2020).

**Table 2.** Chem vs. No-chem pairs chosen to compare chem and no-chem models

| Chem | No-chem |
|---|---|
| CESM2-WACCM | CESM2 |
| CESM2-WACCM-FV2 | CESM2-FV2 |
| GFDL-ESM4 | GFDL-CM4 |
| UKESM1-0-LL | HadGEM3-GC31-LL |
| SOCOLv4 | MPI-ESM1-2-LR |
| GISS-E2-1-G (p3) | GISS-E2-1-G (p1) |
| GISS-E2-2-G (p3) | GISS-E2-2-G (p1) |

## 2.2 CESM-PORT

Parallel Offline Radiative Transfer in the Community Earth System Model (CESM-PORT) (Conley et al., 2013) is driven by model-generated datasets that can be used for any radiation calculation that the underlying radiative transfer schemes can
perform, such as diagnosing radiative forcing. The inclusion of stratospheric temperature adjustment under the assumption of fixed dynamical heating is necessary to compute radiative forcing in addition to the more straightforward instantaneous radiative forcing.

The CESM-PORT experiment includes two parts: verification and perturbation. We validate CESM-PORT using the ozone data from the Whole Atmosphere Community Climate Model (WACCM) (Marsh et al., 2013) under piControl configurations
with interactive chemistry. For the perturbation process, another series of CESM-PORT experiments are carried out using ozone mixing ratio from abrupt-4×$CO_2$ experiment of CMIP6 models with interactive chemistry. This is done by first computing the ozone fraction ratio of abrupt-4×$CO_2$ to piControl, and then multiplying the background ozone mixing ratio of the CESM-PORT piControl experiment above its tropopause by the corresponding fraction ratio obtained for each model, which has been interpolated onto the CESM grid.

The radiative forcing of the surface-troposphere system due to the perturbation or the introduction of an agent is defined as "the change in net (down minus up) irradiance (solar plus long-wave; in $Wm^{-2}$) at the tropopause after allowing for stratospheric temperatures to re-adjust to radiative equilibrium, but with surface and tropospheric temperatures and state held fixed at the unperturbed values" (Ramaswamy et al., 2001). It can be investigated by looking at the difference of radiative fluxes and temperature adjustment between the CESM-PORT reference and perturbation runs.

## 2.3 Pairs of chem and no-chem models

To investigate the impact of interactive chemistry, we compare models that simulate interactive ozone with those that impose a fixed pre-industrial climatological ozone forcing data-set (i.e. the "no-chem" models) from Checa-Garcia et al. (2018), following the method discussed in Morgenstern et al. (2022). We identified seven such "pairs" (Table 2)

# 3 Results

In this section, we examine the response of ozone mixing ratio and column ozone abundance from all the chem models except for GISS-E2-2-G since it has similar behavior with GISS-E2-1-G. We then investigate potential drivers of these responses. The responses are assessed by taking the difference between the climatology obtained from the last 100 years for abrupt-4$\times$CO$_2$ (4$\times$CO$_2$ hereafter) or years 135 to 145 for 1pctCO$_2$ experiment (when it reaches the same CO$_2$ level as abrupt-4$\times$CO$_2$) and the climatology of the 150-year-long piControl experiments.

## 3.1 Annual-mean zonal-mean ozone response

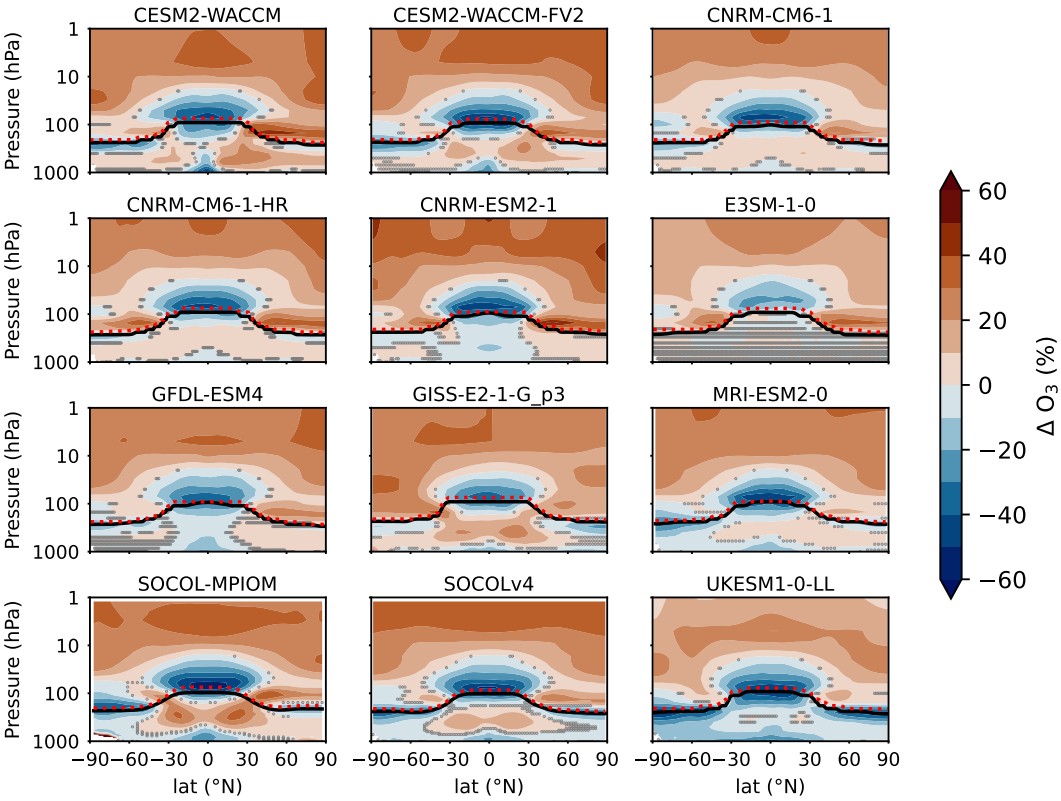

**Figure 1.** Annual-mean zonal-mean ozone response to 4$\times$CO$_2$ of each chem model. Tropopause for piControl and 4$\times$CO$_2$ is denoted using black and red dotted line, respectively. Regions that are not stippled are statistically significant (at 99% level), according to the t-test.

Figure 1 shows the annual-mean zonal-mean ozone response to 4$\times$CO$_2$. We assume the timeseries of ozone concentration under piControl and 4$\times$CO$_2$ are independent samples with the same variance, then we compute the t statistic to see if the two samples have same mean value. This also applies to other variables we analyze hereafter. Ozone increases in the tropical middle troposphere in most models, and decreases in the upper troposphere (although not the case for CNRM-ESM2-1, GFDL-

170 ESM4 and UKESM1-0-LL). The increase is likely due to enhanced $NO_x$ emission from lightning, which can increase ozone abundance by the cycling of $NO_x$ and $HO_x$ (Banerjee et al., 2014; Revell et al., 2015; Iglesias-Suarez et al., 2018), and increased stratosphere-troposphere exchange through isotropic mixing (Abalos et al., 2020; Hegglin and Shepherd, 2009; Wang and Fu, 2023). A similar pattern was simulated in some of the CCMI1 models (Morgenstern et al., 2018), even though not all those models fully represent NOx production changes under climate change. The decrease near the tropical tropopause is linked to
175 the expansion of the tropopause due to tropospheric warming, which replaces ozone-rich stratospheric air with tropospheric air that has a lower ozone abundance. Note that E3SM-1-0 does not employ interactive chemistry in the troposphere, which explains the lack of response in this region for this model. When the $CO_2$ forcing is transient rather than abrupt (1pctCO$_2$), we find a largely similar response (see Fig. B1).

In the stratosphere, the ozone mixing ratio increases in the US due to the reduction in chemical destruction caused by cooling.
There is a decrease in tropical LS ozone of about 40%, which is caused mostly by dynamical processes. But partly, it also results from the increased US ozone that absorbs more ozone-producing UV radiation (200-240 nm), thus less ozone will form in the LS (self-shielding, Dütsch et al., 1991). The relative effects of both processes could be partially isolated by using the $4\times CO_2$ experiment with prescribed sea surface temperatures (SSTs) from the piControl experiment, as has been done in Match and Gerber (2022) and Chrysanthou et al. (2020), suggesting that the self-shielding effect would be responsible for up to one third
of the total, though prescribing the SSTs doesn't fully prevent tropical upwelling from changing. The strengthened transport by the BDC of the tropical ozone to the extratropics and the expansion of the tropopause will also lead to the replacement of ozone-rich air with ozone-poor air. In the extratropics, the enhanced ozone transport from the tropics and upper levels is more significant than the change of the efficiency of ozone formation by photolysis, resulting in a net increase in the LS.

Comparing the ozone response of these models with CMIP5 (Chiodo et al., 2018), we see that the pattern in CMIP6 models
generally agrees well, indicating consistency between the two generations of models. However, having more models for our analysis allows us to test whether the mechanisms of ozone response based on CMIP6 simulations are more robust.

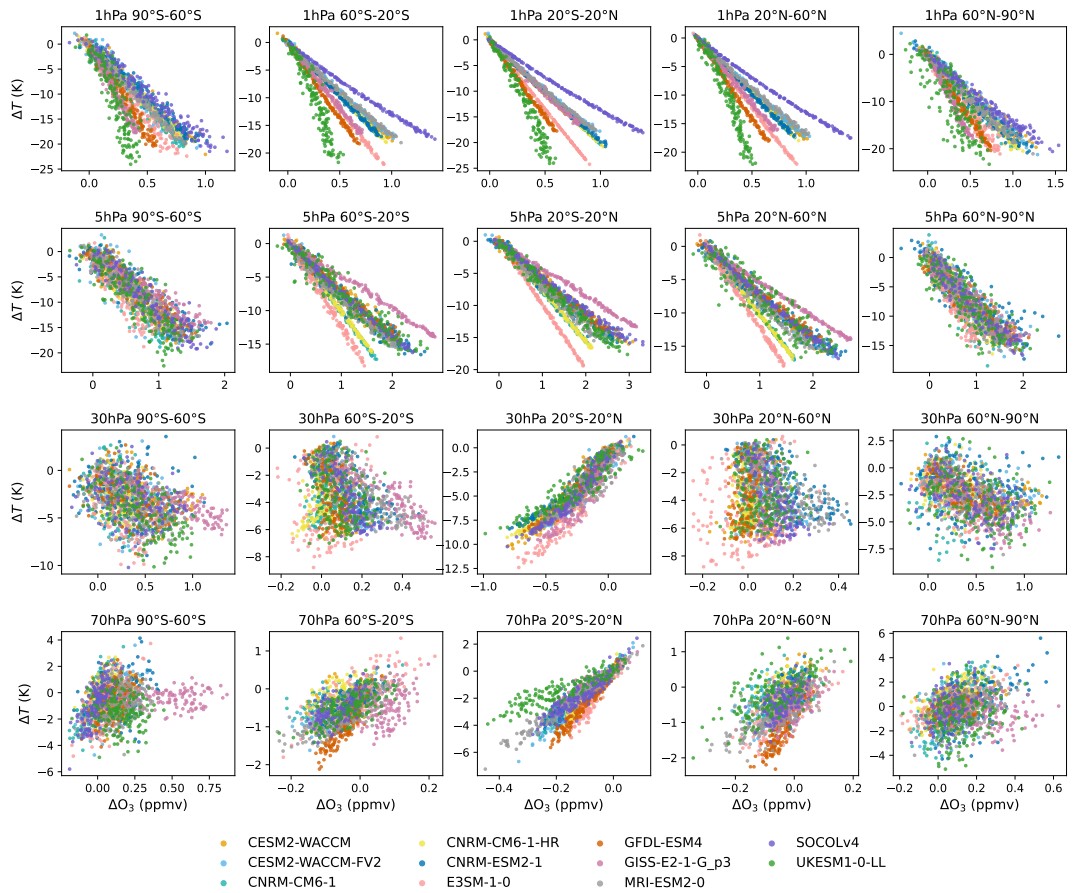

**Figure 2.** 150-year-long annual-mean ozone response to temperature change in stratosphere at different pressure levels and latitude bands based on the 1pctCO$_2$ experiment.

Next, we explore the relationship between ozone and local temperature, to identify how chemistry and transport affect ozone in different stratospheric regions. We look at this relationship by plotting the annual mean temperature change against the ozone response at different pressure levels and latitude bands (Fig. 2). Here, we use the data from 1pctCO$_2$ experiment instead of the

4×CO$_2$ experiment, since the linearly increasing forcing allows tracking the regional dependencies in individual models.

In the US, there is an inverse relationship between the ozone and temperature response for most models: this is expected from the slow-down of ozone destruction with radiative cooling from increased CO$_2$. On the other hand, more ozone enhances UV radiation absorption, heating up the US, and reducing the cooling from increased CO$_2$. The competition between the two processes determines the cooling trend in the US, and differences in the photochemistry schemes of the models explain the

different sensitivities of ozone to temperature change across models, defined by the slope in the scatter plots. For instance, the ozone increase at 1 hPa in SOCOLv4 is the largest by the end of the 1pctCO$_2$ experiment, but it does not correspond to the largest cooling, which might be caused by more efficient heating by ozone. For UKESM1-0-LL, we find the opposite behavior,

indicating a lower temperature-dependence of gas-phase ozone chemistry in this model. The general trend is consistent among all latitude bands at 1 hPa and 5 hPa.

The correlation is the opposite in the tropical LS (30 hPa and 70 hPa) since dynamics and related changes in ozone transport play a dominant role. Here, tropical LS cooling is the result of strengthened upwelling caused by the acceleration of BDC (Abalos et al., 2021). The stronger upwelling results in enhanced transport of ozone out of the tropical pipe, leading (locally) to a decrease (Oman et al., 2010). Hence, the colder the upper troposphere and lower stratosphere (UTLS) gets over the course of the 1pctCO$_2$ experiment, the more ozone is transported out of the tropics, locally reducing ozone abundances (i.e. a positive

relationship). Again, this relationship is model-dependent, and in some models it becomes less linear (e.g. UKESM1-0-LL). In the extratropical LS, the relation is less evident since dynamics and chemistry are equally important.

  Overall, the response of ozone to elevated CO$_2$ is consistent with previous research, and the mechanisms of the responses are examined using CMIP6 data and a larger number of models. The ozone response pattern in the stratosphere can therefore be described as follows: ozone increase in the US dominated by the chemistry response to temperature and ozone heating,

ozone decrease in the tropical LS caused by ozone self-shielding and transport, and increase in extratropical LS as a response of both chemistry and dynamics.

### 3.1.1 Polar vortex vs. ozone response

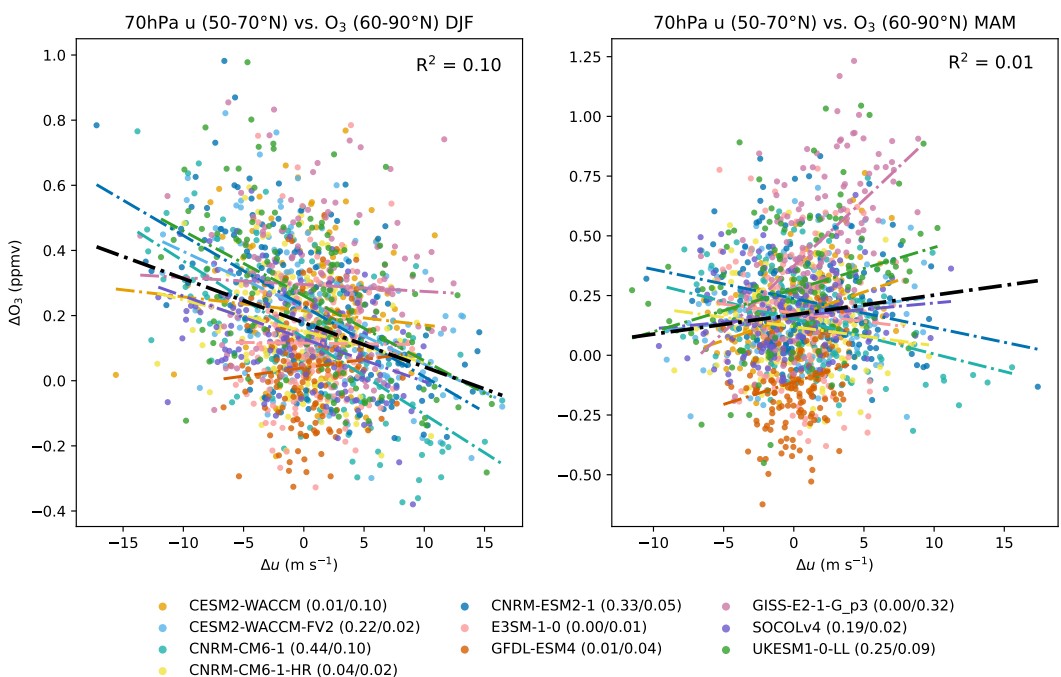

**Figure 3.** 150-year-long seasonal-mean ozone response in 60-90°N to zonal wind ($u$) change in 50-70°N at 70hPa in DJF and MAM for 1pctCO$_2$. Fitting lines retrieved from linear regression are plotted as dash-dotted lines with the corresponding color for each model. The thick black line is fitted using data from all models with the corresponding $R^2$ denoted in the upper right corner of the plot. $R^2$ values for each model are denoted in the legend for DJF and MAM respectively.

Figure 3 shows that in winter, for most models, the weakening of the NH polar vortex reflected by the weakened zonal winds in 50-70°N correlates with an increase of ozone in the Arctic (small but significant negative slope). Antarctic vortex shows
similar behavior (see Figure B2). It might be caused by the more efficient wave forcing from surface warming, and results in enhanced mixing of ozone-rich air masses into the polar vortex. Also, the heating from more ozone could partly contribute to polar warming and weakening of the polar vortex and there is more ozone in the surf zone available to be transported into the polar region. Distinguishing the relative importance of individual factors would require additional experiments, where ozone would respond to dynamical changes but would not in turn affect the radiation and, thus, the circulation. The relation in spring
($R^2 = 0.01$) is not as evident as in winter ($R^2 = 0.10$). The breakup of the polar vortex may lead to enhanced transport of ozone to polar region, but averaging over MAM may mask this relationship. Investigation of the breakup time of polar vortex and how it changes under climate change would need to be considered for each models, which is out of scope, but which merits further investigation. Note that in the 1pctCO$_2$ simulations analyzed here, ODSs are set to pre-industrial values, and heterogeneous chemistry is less important than under present-day conditions. Hence, polar ozone abundances are mostly determined by

mid-winter transport, and thus correlate better with the vortex strength in winter than in spring. Under near-present day ODSs concentrations, the springtime ozone/vortex relationship gets magnified by the combined feedback of heterogeneous chemistry, temperature and dynamics (Kult-Herdin et al., 2023).

### 3.1.2 Tropical upwelling vs. ozone response

In the models used here, surface temperature increases in most regions, with some exceptions over the North Atlantic in some
235 models (see Fig. B2), which will lead to more efficient wave generation and propagation, thus enhancing the upwelling in the tropics, which is consistent with previous studies (Chrysanthou et al., 2020). It is thus pertinent to ask whether the climate model sensitivity correlates with tropical upwelling, and thus ozone. We explore this potential linkage by examining the relationship between upwelling and tropical ozone, using grid-scale vertical velocity ($\overline{w}$, Fig. 4), and residual upwelling ($\overline{w^*}$) diagnosed via the Transformed Eulerian Mean (TEM, see Fig. B3), provided via the DynVar initiative for some of the CMIP6 models (Gerber
and Manzini, 2016). Both metrics indicate strengthened tropical upwelling with warming (panels (a)-(b) in Fig. 4 and Fig. B3), consistent with previous work (Abalos et al., 2021). Again as expected, we find an inverse relationship between upwelling and ozone changes (panels (c)-(d) in Fig. 4 and Fig. B3), indicating that more tropical upwelling decreases tropical ozone. Climatological differences (panels (e)-(f) in Fig. 4 and Fig. B3) are also consistent between the models so that they all show a negative lower-stratospheric tropical ozone change and a positive in tropical upwelling. However, we don't find a consistent
relationship in the inter-model climatological spread of the two variables, which seems to be dominated by "outliers" (e.g. SOCOL-MPIOM in panel (e)) and the regression slope there is insignificant. The sensitivity of ozone to upwelling may differ across models, as it also depends on the vertical ozone gradient, which differs across models (e.g. in UKESM1-0-LL, not shown) and the background climatology of stratospheric transport and dynamics.

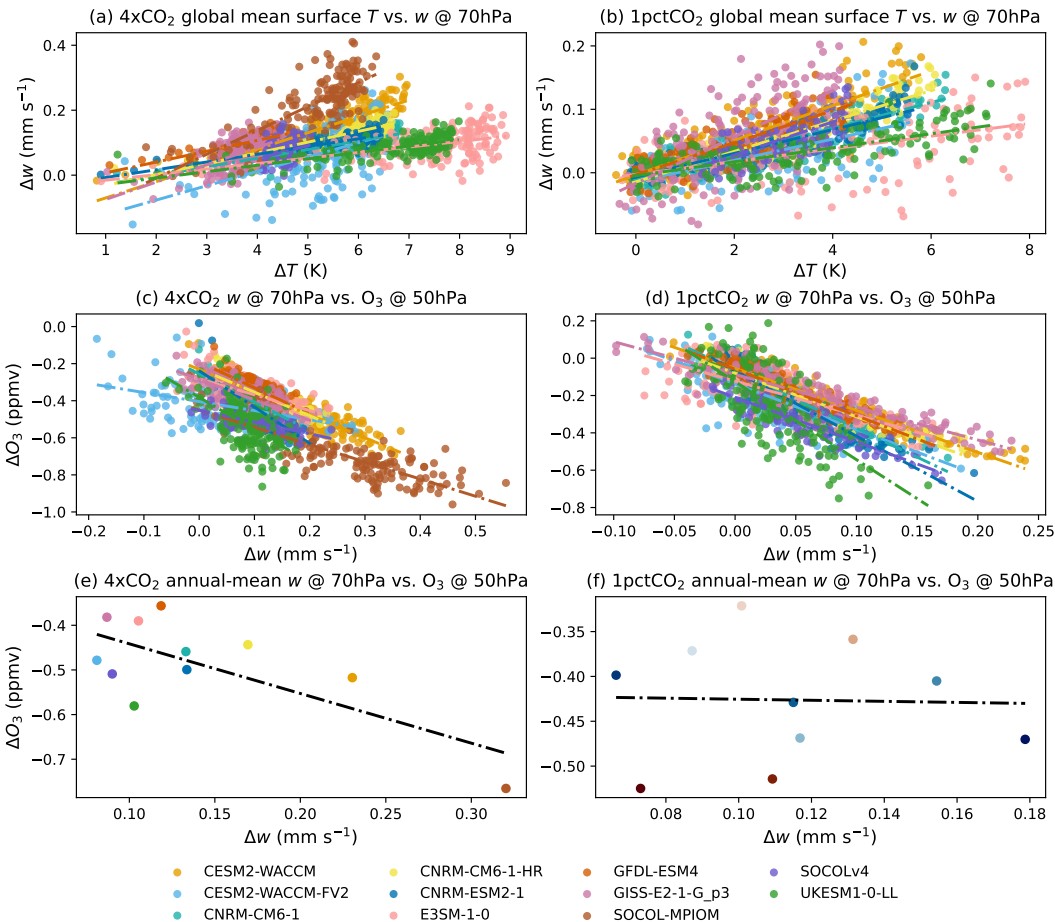

**Figure 4.** (a)-(b): Annual-mean tropical (15°S-15°N) upwelling (w) response to global mean surface temperature for the $4\times CO_2$ and $1pctCO_2$ experiments for individual year. (c)-(d): Same as (a)-(b), but contrasting upwelling changes with the ozone response. (e)-(f): Same as (c)-(d), but averaged over the last 100 years for $4\times CO_2$ and for all the 150 years for $1pctCO_2$ experiment. Fitting lines retrieved from a linear regression are plotted with the corresponding color of each model.

## 3.2 Column ozone response

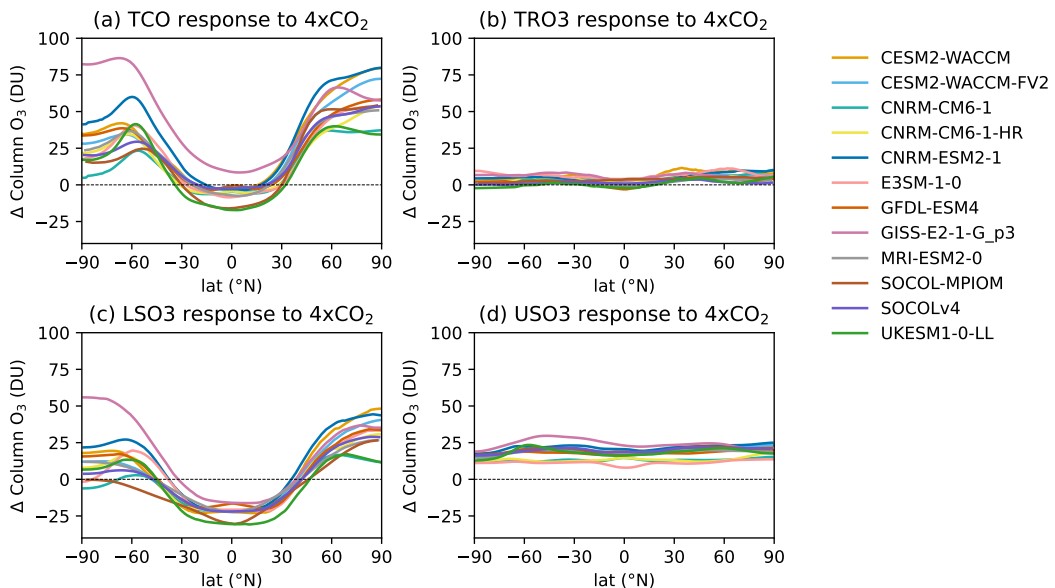

**Figure 5.** Annual-mean column ozone response to $4\times CO_2$ change. (a) TCO, (b) tropospheric (TRO3), (c) lower-stratosphere (LSO3), and (d) upper-stratosphere (USO3) partial ozone columns. The lower stratosphere is defined as the atmospheric layer between the tropopause and 20 hPa, and the upper stratosphere is defined as the layer between 20 hPa and 1 hPa.

One of the key ozone metrics of interest is TCO, as it affects the amount of UV reaching the Earth's surface: this is shown in Figure 5. TCO increases by up to 50-75 DU in the polar regions, while it's around zero in the tropics. In the tropics, the multi-model uncertainty is smaller, though UKESM1-0-LL and SOCOL-MPIOM show a more negative TCO response and GISS-E2-1-G_p3 shows instead a high bias. In the extratropics, the spread among models is increasing and is the strongest in the polar regions, with values ranging between 5 and 80 DU in SH and 30 and 80 DU in NH. Decomposing the response of

TCO into three parts, we see that for TRO3, the response is relatively small and slightly positive in the middle latitudes. The LSO3 response dominates the uncertainty of the model spread in TCO with a negative response in the tropics and a mostly positive but highly model-dependent response in the high latitudes; and for USO3, there is an uniform increase. Therefore, the compensation between LSO3 and USO3 leads to small response in the tropics for TCO. These results are consistent with the analysis of the data from four CMIP5 models (Chiodo et al., 2018), including also the response in the NH being larger than

that in the SH due to a stronger BDC (Butchart, 2014), and are similar with previous study (Morgenstern et al., 2018). When looking at the TCO seasonal cycle, there is a larger seasonal variation at high latitudes than in the tropics with the peak in both hemispheres occurring in the late boreal winter and early spring (not shown). The seasonal variability in high latitudes is consistent with that of BDC, which is stronger in the winter-spring hemisphere (Shepherd, 2008).

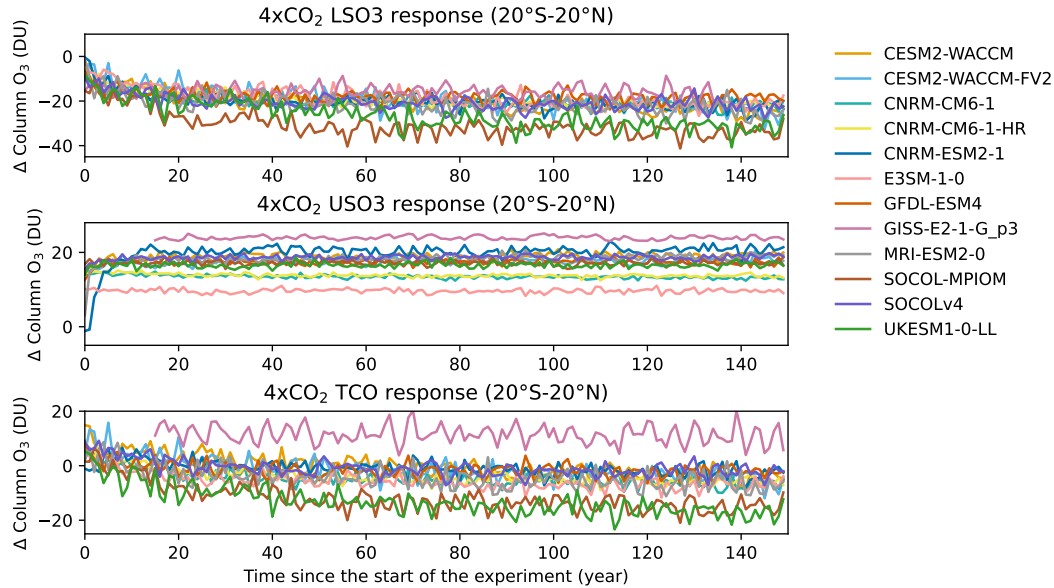

**Figure 6.** 150-year-long annual-mean tropical (20°S-20°N) LSO3, USO3 and TCO response to a $4\times CO_2$ change with time.

Figure 6 shows that tropical LSO3 decreases quickly in the first ~40 years, followed by a slight trend potentially due to
the slow response of the deep ocean, while tropical USO3 reaches equilibrium almost instantaneously. This confirms again the
different dominant mechanisms in the two layers. In the LS, the change in transport dominates, while in the US, the cooling-
induced change in the efficiency of ozone destruction in the Chapman mechanism dominates. Since the response of dynamics
is much slower, as it is linked to the surface temperature changes, it takes longer for LSO3 to reach equilibrium. As a result of
this lag in the LSO3 response, the tropical TCO response to the $4\times CO_2$ increase is slightly positive for the first several decades
and then gets mostly negative over the subsequent 80 years. For 1pctCO$_2$ (see Fig. B4), we can see the lag of the response
since the tropical TCO response around year 140 is smaller than the equilibrated value from $4\times CO_2$. This is also revealed in
the slightly larger decrease of ozone in tropical LS for $4\times CO_2$ (see Fig. 1) compared to 1pctCO$_2$ (see Fig. B1).

## 3.3 Climate feedback

### 3.3.1 Temperature adjustment

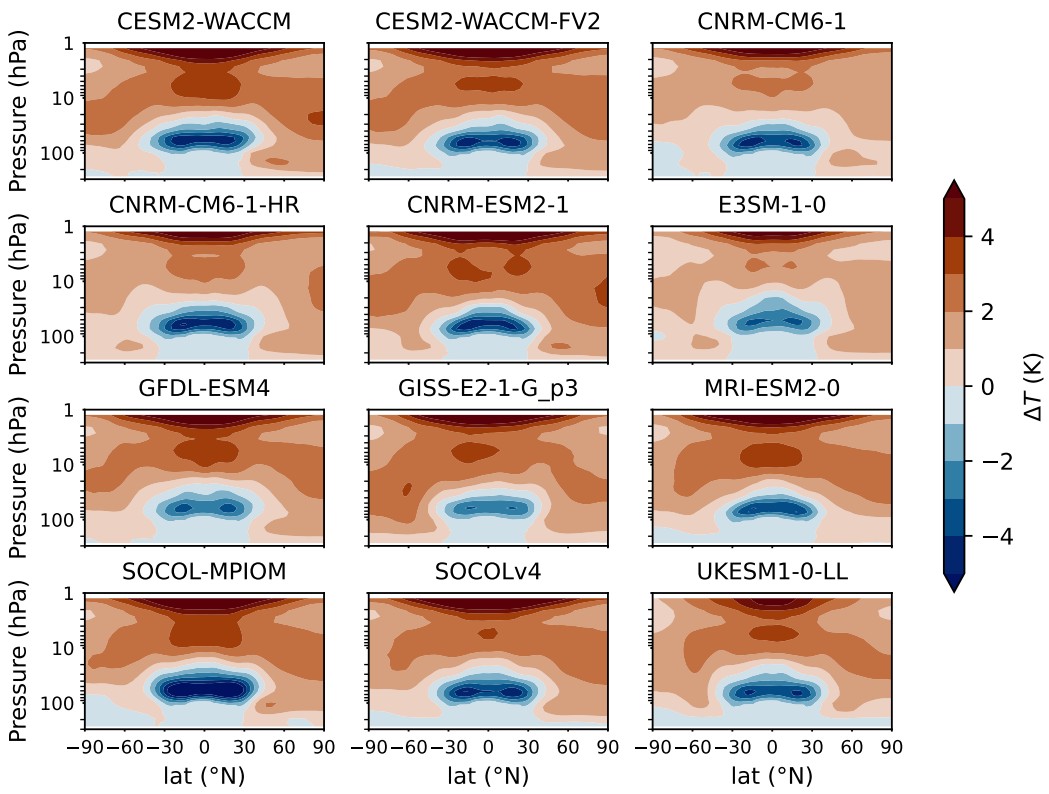

**Figure 7.** Temperature adjustment to ozone for the $4\times CO_2$ experiment from the chem models.

So far, we have examined how ozone is affected by $CO_2$-driven warming, via changes in circulation and temperature. However, ozone also largely affects stratospheric climate, via changes in radiative heating. One way to disentangle the radiative effects of ozone is by performing offline radiative transfer calculations to quantify the temperature adjustments needed to achieve radiative equilibrium, by using the Fixed Dynamical Heating (FDH) approximation with fixed tropospheric temperature (Fels et al., 1980). We achieve this in CESM-PORT for all the ozone perturbations derived from each of the models displayed in Figure 1 (see Section 2.3), and plot the corresponding temperature adjustments in Figure 7. We find that the temperature adjustments broadly correspond to the pattern of ozone responses to $4\times CO_2$ change (Fig. 1), with a cooling in the tropical LS and warming elsewhere.

By comparing the temperature adjustment to ozone with the actual zonal-mean temperature response in the coupled experiments (displayed in Fig. B5), we can see the relative contribution of the radiative heating induced by ozone to stratospheric temperature changes. In the LS, we find a slight warming in the NH middle and high-latitudes, contributing to about 25% of the

total temperature response. Over the tropical tropopause and LS, we find cooling, indicating that reduced ozone in the UTLS amplifies the stratospheric cooling from increased $CO_2$ and can explain about half of the total temperature change, whereas in the US, heating from increased ozone is outweighed by radiative cooling from increased $CO_2$, consistent with previous work on historical temperature trends (Chiodo and Polvani, 2022; McLandress et al., 2011). At high latitudes in the LS, the temperature response depends on the opposing influences of warming induced by increased ozone abundances and downwelling, and radiative cooling from $CO_2$ (Chiodo et al., 2023; Kult-Herdin et al., 2023). For most of the models, the warming dominates at high latitudes, while the cooling dominates close to the tropics.

### 3.3.2 Radiative impacts of ozone

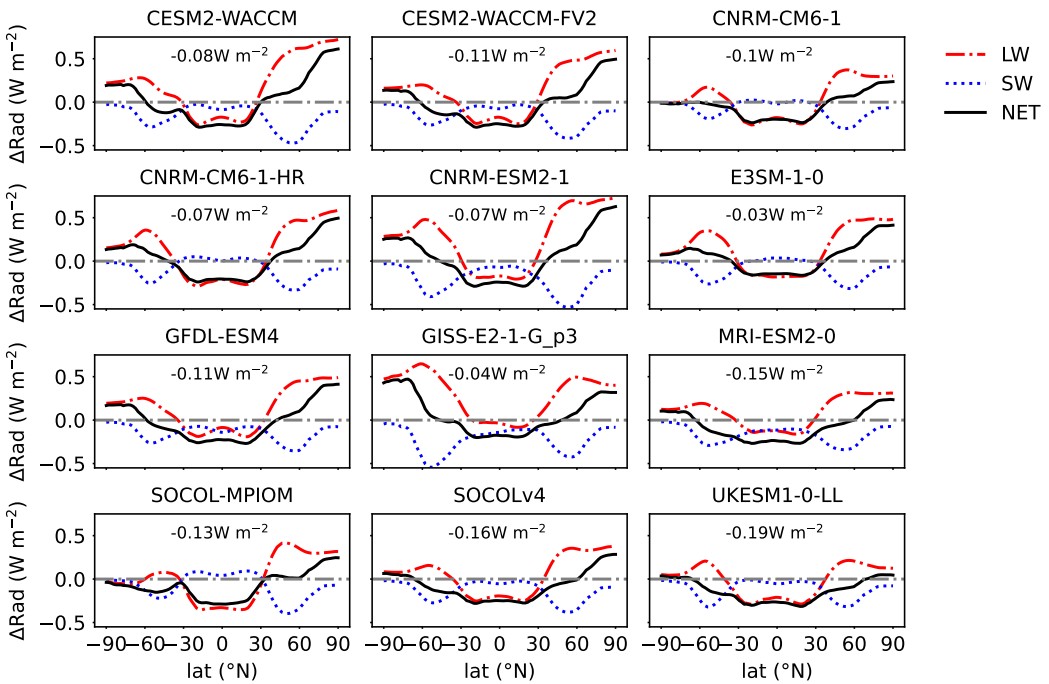

**Figure 8.** Annual-mean zonal-mean response of radiative fluxes at the tropopause to ozone from chem models for the $4\times CO_2$ experiment. The longwave, shortwave and net radiative flux response are denoted with red (dash-dotted), blue (dotted) and black lines, respectively. The value of the global mean net flux change is shown as number in each subplot.

Changes in ozone can have a sizable impact on the radiative balance of the stratosphere, via changes in shortwave (SW) heating, but also in the longwave (LW), by changing the trapping of LW radiation coming from the troposphere, and via the stratospheric temperature adjustments. Hence, changes in ozone that are induced by $CO_2$ lead to a radiative forcing (RF), potentially altering tropospheric and surface climate response to $CO_2$ in some models (Dietmüller et al., 2014; Muthers et al., 2014; Nowack et al., 2015, 2018). We quantify the ozone-induced RF across all CMIP6 models, by calculating the stratospherically-adjusted

changes in the LW and SW at the tropopause. By inspecting these quantities in Figure 8, we see that the latitudinal structure of the LW flux changes largely corresponds to the temperature adjustment in the LS and is consistent with the ozone changes near the tropopause. The net LW flux is reduced (meaning more LW escaping the troposphere) in the tropics, and is increased in the high latitudes. These opposing effects are due to changes in the LW absorption as well as in the local temperatures induced by ozone. The reduced ozone abundances near the UTLS radiatively cool the UTLS, leading to reduced LW emission from the stratosphere towards the troposphere; conversely, increased ozone abundances in the LS at high latitudes lead to warming, increasing the LW emission from the stratosphere towards the troposphere. In the SW range, the resulting RF mirrors the changes in TCO and is due to the "shielding" effect of the ozone columns; the SW flux barely changes in the tropics (due to small changes in TCO there), while it's reduced in the extratropics (where TCO increases). At high latitudes, the thicker TCO absorbs more SW, reducing the SW flux reaching the tropopause. The net flux change is the sum of that of LW and SW, and is negative in the tropics and positive in the extratropics. Most importantly, the LW generally dominates over the SW forcing, leading to a global mean negative net RF in all models, varying between -0.03 (E3SM-1-0) and -0.19 Wm$^{-2}$ (UKESM-1-0-LL). Compared to CMIP5, the range in the ozone-induced RF is larger (Fig. 2, e.g. Chiodo and Polvani, 2019) in CMIP6, possibly due to the larger number of models considered in this study. Taken together, the negative RF at the tropopause implies a reduction in tropospheric and surface warming from ozone, consistent with some previous studies (Dietmüller et al., 2014; Muthers et al., 2014; Nowack et al., 2015). However, the ozone-induced RF does not tell the full story, as it does not consider other physical feedbacks or large-scale circulation changes, as well as it can be dependent on the background model climatologies, while in the PORT calculations we used a single prescribed background. Therefore, we examine the overall feedback by comparing models with and without interactive chemistry in the next section.

### 3.3.3 Impact of interactive ozone chemistry on the coupled response

In this section, our aim is to examine the overall climate feedback in the context of coupled experiments. The cleanest way to achieve this is by running pairs of experiments within the same model system, with and without interactive ozone, as done in previous work (Dietmüller et al., 2014; Muthers et al., 2014; Nowack et al., 2015; Chiodo and Polvani, 2016; Marsh et al., 2016; Chiodo and Polvani, 2019). As an alternative approach, we compare the seven pairs of chem and no-chem models from CMIP6 (see Table 2), and look at the differences in the $4\times CO_2$ response among them. The caveat is that the comparison of such pairs will not only isolate the impact of interactive ozone, but also other effects, such as differences in the model physics, as discussed in Morgenstern et al. (2022). More specifically, the pairs also differ in other components like gravity wave drag, and convective parameterization.

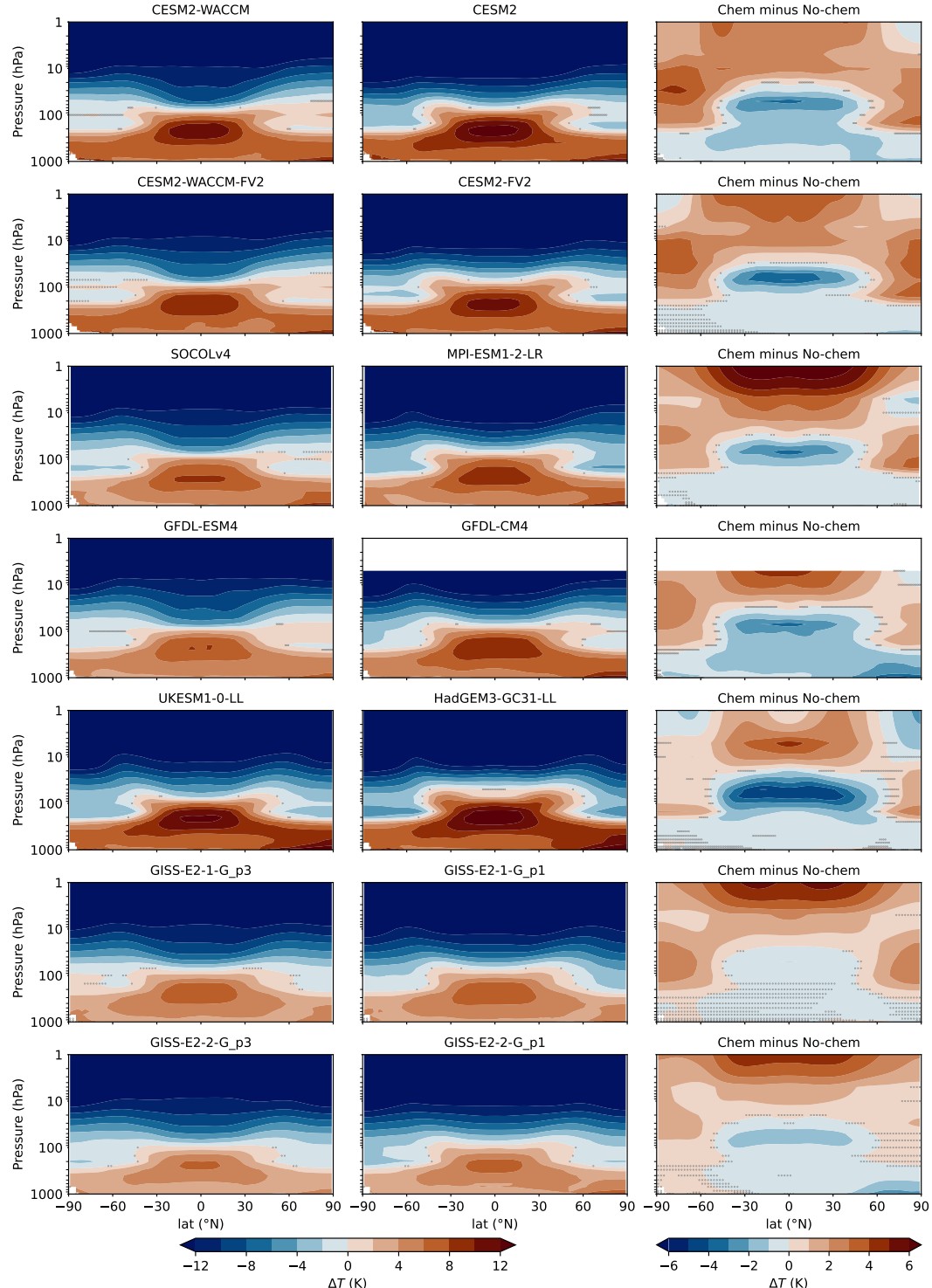

**Figure 9.** Comparison of the 100-year long annual-mean air temperature response to 4×CO₂ between seven pairs of chem and no-chem CMIP6 models. The left column shows the response from chem models, the middle column shows the response from no-chem models and the right column shows the difference between chem and no-chem models.

First, we start by looking at zonal-mean temperature in Figure 9. The seven pairs of models share the same pattern of zonal-mean temperature response. Compared with no-chem models, chem models have significantly less warming in the troposphere, more cooling in the tropical LS and less cooling in the extratropical LS and US, which is consistent with previous findings (Dietmüller et al., 2014; Nowack et al., 2015; Marsh et al., 2016). The negative stratospherically-adjusted RF at the tropopause (Fig. 8) might partly explain the reduced tropospheric warming in chem models. Another coupling mechanism could be related to the transport of water vapor from the upper troposphere to the LS being reduced due to the cooling of LS, which would lead to less tropospheric warming due to the GHG effects of the water vapor (Nowack et al., 2018; Banerjee et al., 2019; Nowack et al., 2023). In the stratosphere, the temperature pattern is coherent with the ozone response, with decreased ozone in the tropical LS leading to cooling and increased ozone in the US and extratropical LS leading to a warming (Fig. 1). Among these pairs, UKESM1-0-LL/HadGEM3-GC31-LL exhibits the largest difference in the tropical LS, while SOCOLv4/MPI-ESM1-2-LR has the largest difference in the US. In the multi-model mean (Fig. 10), chem models are about $\sim$2 K cooler in the troposphere in their 4$\times$CO$_2$ response, $\sim$3 K cooler in the tropical LS and $\sim$4 K warmer in the US. In the stratosphere, the temperature pattern is broadly consistent with the temperature adjustments calculated with CESM-PORT (Fig. 7), suggesting that the chem vs no-chem differences are indeed indicative of a true "ozone effect". These temperature changes have implications for the zonal mean zonal wind, as discussed in the next paragraph. In the tropical stratosphere, we explore if the inclusion of interactive ozone also affects the degree of BDC acceleration, as suggested by (Hufnagl et al., 2023). Out of the three pairs of experiments that provide tropical upwelling ($\overline{w^*}$, Table B1), for all of them we find a reduction in the increase due to CO$_2$ in the chem version of the models. Changes in tropical upwelling critically depend on zonal wind changes near the tropical tropopause layer (TTL), as these modulate the propagation and dissipation of tropospheric waves. Ozone decreases the meridional temperature gradient near the TTL, altering the BDC as a consequence of changes in zonal winds, as discussed next.

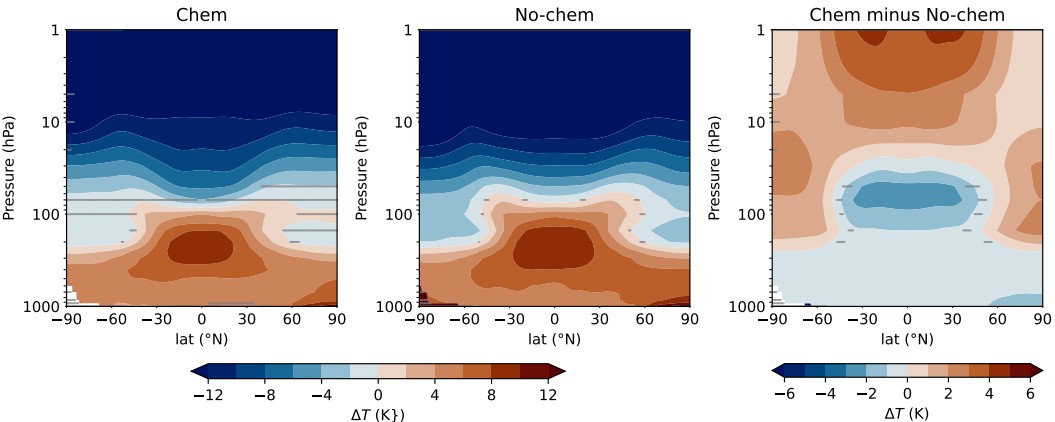

**Figure 10.** Similar to Figure 9, but average of 100-year long annual mean air temperature response to 4$\times$CO$_2$ of seven pairs of chem and no-chem models.

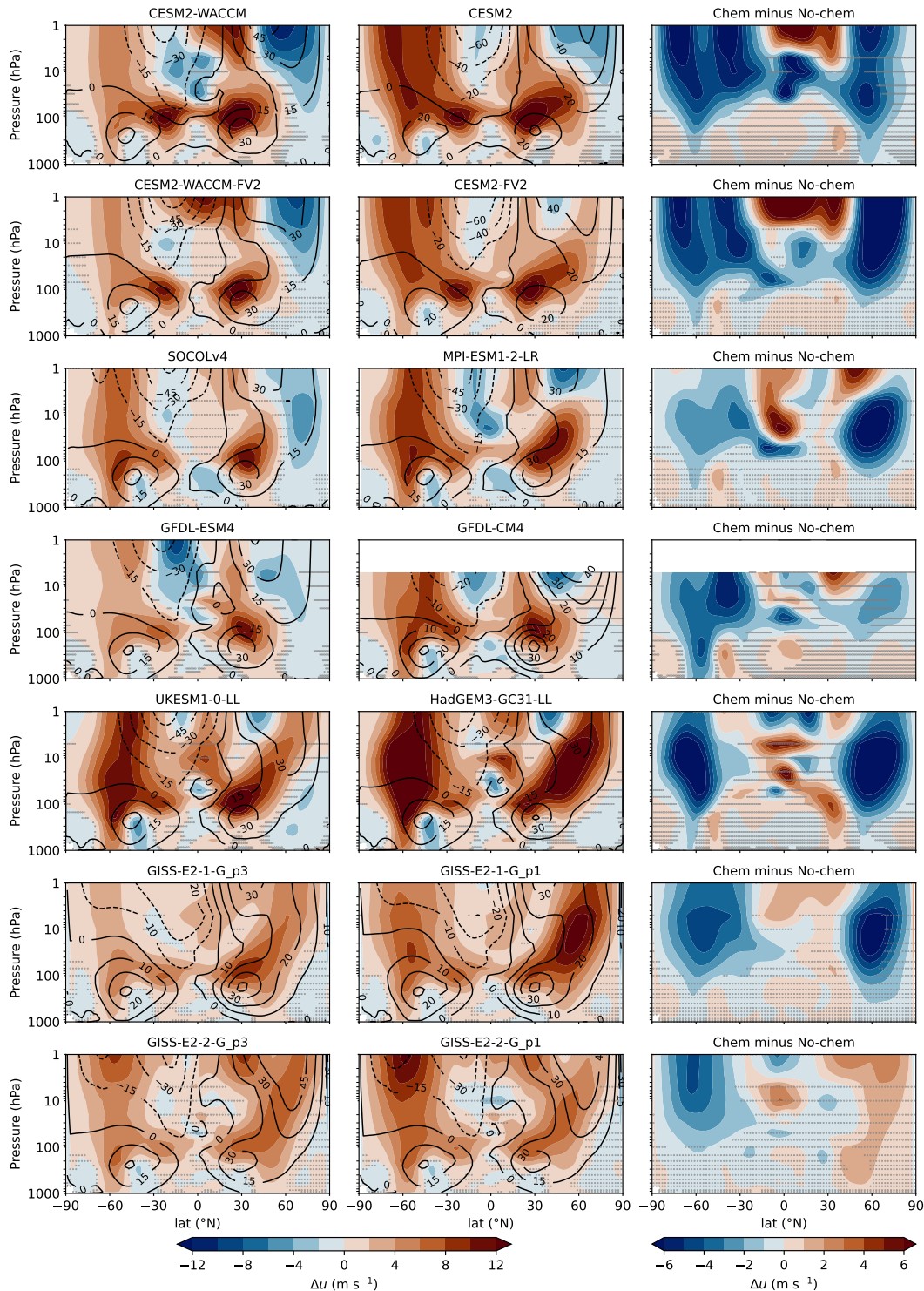

**Figure 11.** Comparison of the 100-year long seasonal mean zonal wind response to a $4 \times CO_2$ change in DJF between seven pairs of chem and no-chem models. Black contour lines depict the zonal wind climatology from the piControl experiment.

We then look at the response of polar vortex in boreal winter (Fig. 11) since this is the season when the strength of the polar vortex is at its peak. All the individual pair comparisons of zonal wind in DJF reveal a weaker polar vortex compared to no-chem models (Fig.11) because chem models have a smaller latitudinal temperature gradient (Fig. 9). The one exception is the GISS-E2-2-G model, in which interactive composition produces a more dramatic weakening of the Atlantic Meridional Overturning Circulation (AMOC); this, on longer timescales, accelerates the NH jet, obfuscating the initial weakening of the NH polar vortex (Orbe et al., 2024). On shorter timescales, however, the initial response of the polar vortex in the interactive chemistry simulation is consistent with the response observed in the other models. In terms of the frequency of sudden stratospheric warmings (SSWs), models widely differ in terms of their background (piControl) SSW frequencies, as well as their effects of global warming, consistent with previous work (Ayarzagüena et al., 2020). Interestingly, chem models tend to show larger and more consistent (4 out of 5 pairs) increases in the SSW frequency compared to no-chem models (Table. B1), in agreement with the zonal wind weakening (Fig. 11). In some models, the non-chem configuration has different sign in the frequency change under abrupt-4xCO2 compared to the chem counterpart (e.g. SOCOLv4/MPI-ESM1-1-2-LR and UKESM1-0-LL/HadGEM3-GC31-LL). In some pairs, the easterly wind anomalies extend to the surface (CESM2-WACCM/CESM2, GFDL-ESM4/GFDL-CM4 and UKESM1-0-LL/HadGEM3-GC31-LL). The strengthening of zonal wind in the tropical US is due to the expansion of the weakened polar vortex as discussed in Section 3.1.1. Averaged over all pairs (Fig. 12), we find that zonal wind weakens in both hemispheres in chem models with respect to non-chem models. Most remarkably, these anomalies extend to the troposphere in the SH, and are indicative of an equatorward shift of the middle latitude jet, thus opposing the effect of increasing $CO_2$. This result is consistent with previous work using individual models, suggesting that models with prescribed ozone may over-estimate the tropospheric circulation response to $CO_2$ (Chiodo and Polvani, 2017; Nowack et al., 2018; Li and Newman, 2023). The crucial new addition is that here, we confirm this finding with more models, and find that it is a robust feature among all models in the stratosphere. This effect could be one of the reasons for uncertainty across some of the CMIP6 models in terms of NH polar vortex response to future $CO_2$ scenarios (Karpechko et al., 2022, 2024). In addition, we confirm the finding of Chiodo and Polvani (2019) concerning the role of stratospheric ozone in reducing the poleward shift of the tropospheric mid-latitude jets, as we show in Figure B6 as a difference between the chem and no-chem multi-model mean pairs in the 850 hPa zonal winds.

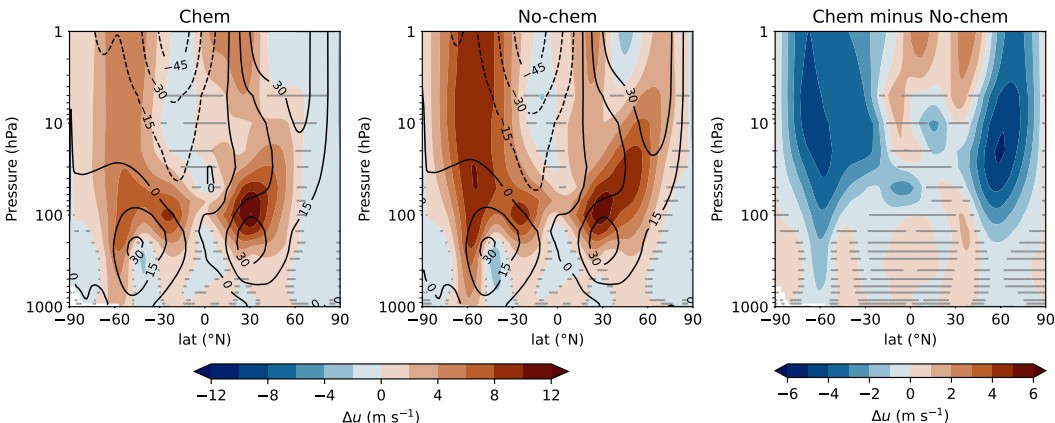

**Figure 12.** Similar to Figure 11, but a model average.

Lastly, we examine the differences in the global-mean surface air temperature response in $4\times CO_2$ and $1pctCO_2$ experiments as well as the difference of the two (see Table 3). Chem models tend to exhibit less surface warming than no-chem models in both types of $CO_2$ experiments, consistent with the tropospheric cooling shown in Figure 9. In the multi-model mean, we find
that chem models have approx. 9.0% less warming under $4\times CO_2$ than no-chem models, while for $1pctCO_2$, the difference is smaller ($\sim$7.0%). This leads to an overall larger response of global mean surface temperature for chem version than no-chem version. Taken together, the negative RF induced by ozone is consistent with the reduction in surface warming in chem models, which is in agreement with the initial work by Dietmüller et al. (2014) and in qualitative agreement with Nowack et al. (2015). However, we still also report a large uncertainty across models in this effect, partly also confirming the previous
difference between the single-model studies (Dietmüller et al., 2014; Nowack et al., 2015; Muthers et al., 2016; Marsh et al., 2016; Chiodo and Polvani, 2019). Also, models with a large RF from ozone (e.g. UKESM-1-0-LL and SOCOLv4) are not the models showing the largest reduction in surface warming, suggesting that other factors such as differences in model physics and other climate feedbacks (e.g. clouds and water vapor) may contribute to the differences between both pairs of models. The bias arising from inconsistencies between the thermal and chemical tropopauses has been eliminated in the HadGEM3-GC31-LL
$4\times CO_2$ and $1pctCO_2$ experiments, as described in Hardiman et al. (2019). Hence, the UKESM-1-0-LL/HadGEM3-GC31-LL pair is not affected by this, but it may be of importance in other pairs.

**Table 3.** Comparison of the global-mean surface air temperature of the seven chem/no-chem CMIP6 model pairs.

| Pair | $4{\times}CO_2$ (K) | | | $1pctCO_2$ (K) | | |
|---|---|---|---|---|---|---|
| | Chem | No-chem | Chem minus No-chem | Chem | No-chem | Chem minus No-chem |
| **CESM2-WACCM/CESM2** | 6.07 | 6.84 | -0.77 | 5.03 | 5.49 | -0.46 |
| **CESM2-WACCM-FV2/CESM2-FV2** | 5.65 | 6.02 | -0.37 | 4.96 | 4.96 | -0.0015 |
| **GFDL-ESM4/GFDL-CM4** | 4.12 | 5.48 | -1.36 | 3.78 | 5.06 | -1.29 |
| **GISS-E2-1-G (r1i1p3f1)/GISS-E2-1-G (r1i1p1f1)** | 3.63 | 3.81 | -0.18 | 3.68 | 3.70 | -0.0167 |
| **GISS-E2-2-G (r1i1p3f1)/GISS-E2-2-G (r1i1p1f1)** | 3.15 | 3.66 | -0.51 | - | - | - |
| **UKESM1-0-LL/HadGEM3-GC31-LL** | 7.3 | 7.47 | -0.17 | 6.54 | 6.64 | -0.10 |
| **SOCOLv4/MPI-ESM1-2-LR** | 4.46 | 4.74 | -0.27 | 4.05 | 4.3 | -0.26 |
| **Multi-model Mean** | 4.91 | 5.43 | -0.52±0.39 (7.2%) | 4.67 | 5.03 | -0.35±0.45 (8.9%) |

## 4 Conclusions

In this study, we investigate the ozone response to elevated $CO_2$ levels, by analyzing data from 20 models from the CMIP6 DECK experiments. We assess the role of potential drivers of ozone changes, by exploring the relationships between ozone and parameters like near-surface and stratospheric temperature, zonal wind, and residual upwelling ($w^*$). We find that most stratospheric ozone changes can be explained by these drivers, but we also find large inter-model differences in the ozone response in some regions, that cannot be explained by any of them. The larger number of models enables us to have a more robust comparison than previous studies, which employed three models at most (Chiodo and Polvani, 2019).

The main findings of this paper can be concluded as follows:

1. The ozone response to $4\times CO_2$ and $1pctCO_2$ is very similar, although the ozone decrease in the tropical lower stratosphere is smaller in $1pctCO_2$ due to the lag of response to transient forcing.

2. The analyzed models exhibit a broadly similar pattern of zonal-mean ozone response, with an increase in the upper stratosphere and extratropical lower stratosphere and a decrease in the tropical lower stratosphere.

3. The ozone response in the upper stratosphere is dominated by changes in gas-phase chemistry, while in the lower stratosphere, chemistry and transport changes both play a role. Therefore, the timescale of the response in the lower stratosphere and upper stratosphere is different. In the upper stratosphere, it reaches equilibrium almost instantaneously because of the quick response of gas-phase chemistry. In the lower stratosphere, the response is much slower because of the slower chemical time-scales, as well as the sizable role of tropical upwelling, which is influenced by SSTs and thus slow equilibration timescale of the ocean.

4. The decrease of tropical lower stratospheric ozone is caused by stronger upwelling, and the increase of Arctic ozone is partly due to weaker westerlies, and thus more in-mixing of ozone-rich air into the polar region.

5. The total column ozone response is negligible in the tropics because of the cancellation between decreases in the lower stratosphere and increases in the upper stratosphere. Total column ozone increases at high latitudes, but with a large inter-model discrepancy compared to the tropics, which is related to the different response of zonal wind in the models. Overall, the pattern (and its uncertainty) in the total column ozone response is dominated by lower stratospheric ozone.

6. The response of upwelling to surface warming, as well as the ozone response to strengthened upwelling, is strongly model dependent. These two sensitivities combined determine how tropical total column ozone responds to increased $CO_2$ concentrations in different models. Disentangling the two effects for ozone response uncertainty would require additional experiments with prescribed SSTs, to constrain the tropical upwelling, and having ozone calculated online but radiatively decoupled.

7. Models with interactive chemistry show less warming under increased $CO_2$ in the troposphere and tropical lower stratosphere, and more warming in the extratropical lower and upper stratosphere, consistent with previous studies. As a

consequence of these temperature changes, we find a weakening of the stratospheric polar vortex during boreal winter under increased $CO_2$; this signal extends to the troposphere. Also, due to the weakening in the polar vortex, chem models tend to show larger and more consistent increases in the SSW frequency compared to no-chem models.

8. Models with interactive chemistry simulate, on average, have about $\sim$9.6%$\pm$7.2% smaller surface warming than models without chemistry under $4\times CO_2$, and about $\sim$7.0%$\pm$8.9% less warming under 1pctCO2.

Previous studies also show that the ozone response to a $4\times CO_2$ change has a considerable impact on the tropospheric circulation in NH, and will induce an equatorward shift of the North Atlantic jet during boreal winter (Chiodo and Polvani, 2019). This shift of the North Atlantic jet may induce a rapid weakening of the AMOC, and in turn, might result in an eastward acceleration and poleward shift of the Atlantic jet (Orbe et al., 2024). Although this is out of the scope of this work, it emphasizes the extensive effect that the ozone response may have and thus the importance of including ozone interactive chemistry in climate sensitivity studies. Therefore, it might worth using the ozone field simulated in chem models as the forcing for no-chem models in future model intercomparison projects such as CMIP7.

A caveat of this work is that the abundance of ODSs in the three experiments is fixed at pre-industrial level to separate the effect of $CO_2$, thus the effect of anthropogenic halogens cannot be simulated. The present-day ODSs level is high, which will lead to ozone depletion through heterogeneous chemistry in polar stratospheric clouds (PSCs) with $CO_2$ induced stratospheric cooling. This may counteract the positive ozone response from a strengthened BDC and weakened westerlies in polar region, resulting in a smaller ozone response in high latitudes. However, we fix the ODSs level in this work following the standard approach of studying climate feedback (Gregory and Webb, 2008; Andrews et al., 2012). Future analyses are needed to study the ozone response considering present-day and future ODSs levels.

The negative (damping) climate feedback from stratospheric ozone changes is in agreement with previous single model studies (Dietmüller et al., 2014; Nowack et al., 2015) utilizing the current available data from CMIP6. Although three out of the seven pairs we chose to conduct the chem/no-chem comparison have other differences other than chemistry such as height of model top, they share similar patterns with those that only differ in chemistry scheme. This indicates that the different chemistry scheme contributes the most to the chem/no-chem comparison and other model differences play a minor role. However, to isolate the feedback of ozone responses, future experiments which compare the same model system with and without interactive chemistry directly should be included in future model intercomparison projects such as CMIP7. Lastly, the large difference of global mean surface air temperature between chem and no-chem models of $\sim$10% cannot be explained by the radiative effect of ozone response alone ($\sim$-0.1 W m$^{-2}$). Other feedbacks, such as via cloud and water vapor changes, as well as biases induced by, e.g., inconsistencies between the chemical and thermal tropopause, might contribute: further studies are needed to explain the cause of these large differences.

*Code and data availability.* CMIP6 model datasets used in this study are available through the Earth System Grid Federation (ESGF; https://esgf-index1.ceda.ac.uk/projects/cmip6-ceda/). Data and code to reproduce the figures in this work can be found on https://doi.org/10.5281/zenodo.14545386.

## Appendix A

Before analyzing the data, some pre-processing processes are necessary for the cohesion of data format and computation of required variables.

### A1  Calculation of tropopause

Since the variable for tropopause height is not available for all models in all three experiments, we compute the tropopause height following WMO definition[1]. The 19 pressure levels of CMIP6 models are not sufficient to compute the tropopause height, thus we first interpolate the zonal-mean annual-mean geopotential height and air temperature to 40 pressure levels to get a better vertical resolution. Then, the annual-mean zonal-mean tropopause height in Pa is calculated. The annual-mean tropopause height for a time period is derived by averaging the annual-mean values. Note that for the abrupt-4$\times$CO$_2$ experiment, we only average over the last 100 years to make sure the system reaches equilibrium, and for 1pctCO$_2$, the time average is done from year 135 to year 145, which is the time series selected around the year 140 when the CO$_2$ concentration reaches 4$\times$CO$_2$.

### A2  Remap of MRI-ESM2-0

The horizontal resolution of air temperature of MRI-ESM2-0 is remapped so that it is the same as that of its ozone mixing ratio.

### A3  Vertical wind velocity for UKESM1-0-LL and GFDL-ESM4

Since the vertical wind velocity in Pa s$^{-1}$ (wap) variable for UKESM1-0-LL piControl is not available, We average over the first two decades (1850-1870) from historical runs of five ensemble members (r9-r13) to represent the equilibrium state of piControl experiment.

Similarly, since the vertical velocity of the residual mean meridional circulation ($\overline{w^*}$) variable for GFDL-ESM4 piControl is not available, We average over the first three decades (1850-1880) from historical run to represent the equilibrium state of piControl experiment.

---

[1]It is defined as the lowest level at which the lapse rate decreases to 2k km$^{-1}$ or less, provided also that the average lapse rate between this level and all higher levels within 2 km does not exceed 2K $^{-1}$.

## A4 Conversion from $\omega$ to $w$

When analyzing vertical velocity, the only available variable is omega in Pa s$^{-1}$, thus to better represent the vertical motion, we convert $\omega$ to $w$ in mm s$^{-1}$ using Eq.(A1):

$$w_j = -1000\omega_j \frac{H}{P_j} \tag{A1}$$

In which $w_j$ is the vertical velocity in mm s$^{-1}$ on j-th pressure level; $\omega_j$ is the vertical velocity in Pa s$^{-1}$ on j-th pressure level; $H$ is the scale height and $P_j$ is the pressure of j-th pressure level.

## A5 Computation of $\overline{w^*}$

Transferred-Eulerian-Mean (TEM) circulation combines the contribution of eddy and mean transport (Butler et al., 2016) and group the eddy fluxes of heat and momentum into the zonal momentum equation (David G. Andrews, 1987). It keeps the benefit of the Eulerian view, but also includes eddy fluxes to understand particles transport from the Langrangian view, and it is usually used in stratosphere.

Since $w^*$ for SOCOL-MPIOM is not provided, we compute them following Eq.(A2):

$$\overline{w^*} = \overline{w} + (a\cos\phi)^{-1}\frac{\partial(\cos\phi\overline{v'\theta'}/(\partial\overline{\theta}/\partial z))}{\partial\phi} \tag{A2}$$

In which $\overline{w^*}$ is the mean vertical velocity of the residual mean meridional circulation, $\overline{w}$ is the mean vertical velocity, $a$ is the mean radius of Earth, $\phi$ is latitude, $v'$ is the deviation of meridional velocity from zonal mean value, $\overline{\theta}$ is the zonal mean potential temperature and $\theta'$ is the deviation of potential temperature from zonal mean value.

## A6 Computation of global-mean surface temp

The global mean surface temperature is computed using weighted mean of surface temperature as Eq.(A3):

$$T_{glbm} = \sum_{i=1}^{i=N} \sigma_i T_i \tag{A3}$$

In which, $T_{glbm}$ is the global mean surface temperature, $N$ is the number of grid cells, $\sigma_i$ is the area of the i-th grid cell, and $T_i$ is the average surface temperature of the i-th grid cell, which is represented by the temperature at the corresponding grid point.

## A7 Computation of column ozone

Column ozone is computed by integrating the number of ozone molecule between certain pressure levels, and then converting it to Dobson unit (DU) following Eq.(A4):

$$Col_{ozone} = \frac{1}{2.69 \times 10^{16}} \int_{P_1}^{P_2} \frac{10 vmr}{M_{air} g} dP \tag{A4}$$

In which, $Col_{ozone}$ is the column ozone value in DU, $P_1$ and $P_2$ are the starting and ending pressure level in Pa, $vmr$ is the
ozone volume mixing ratio, $M_{air}$ is the mass of one air molecule in gram.

## A8  Comparison of chem and no-chem models

For the comparison of chem and no-chem models, We first remap all models' horizontal grid to the same resolution, which is
chosen as 288×192. Since all CMIP6 models have 19 pressure levels, vertical interpolation is done on SOCOLv4 so that the
data are on the same pressure levels.

## Appendix B

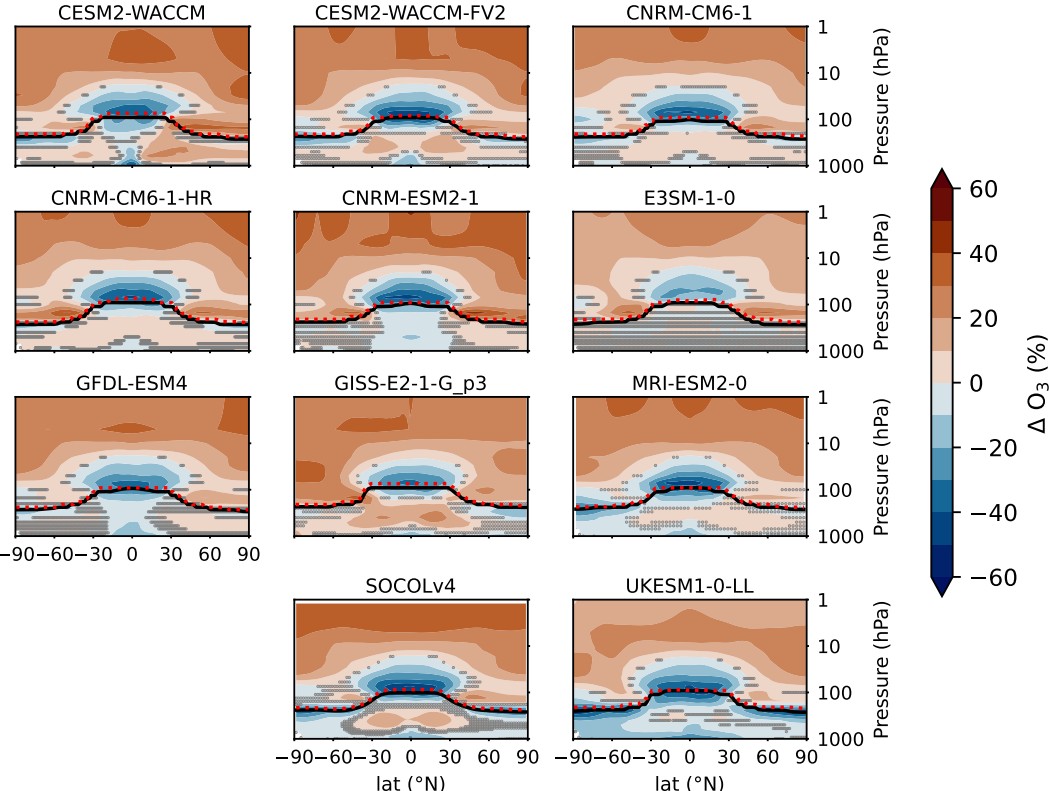

**Figure B1.** Annual-mean ozone response to 1pctCO$_2$ of each chem model. Tropopause for piControl (1pctCO$_2$) is denoted using black (red
dotted) curve. Regions that are not stippled are statistically significant (at the 99% level), according to the t test.

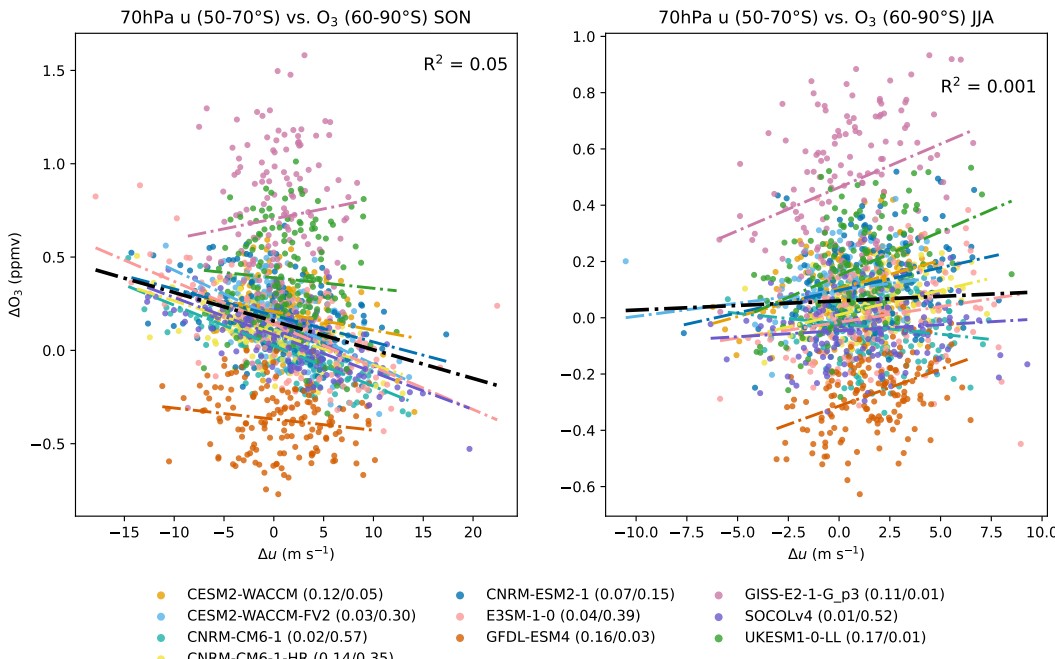

**Figure B2.** 150-year-long seasonal-mean ozone response in 60-90°S to zonal wind ($u$) change in 50-70°S at 70hPa in SON and JJA for 1pctCO$_2$. Fitting lines retrieved from linear regression are plotted as dash-dotted lines with the corresponding color for each model. The thick black line is fitted using data from all models with the corresponding R$^2$ denoted in the upper right corner of the plot. R$^2$ values for each model are denoted in the legend for SON and JJA respectively.

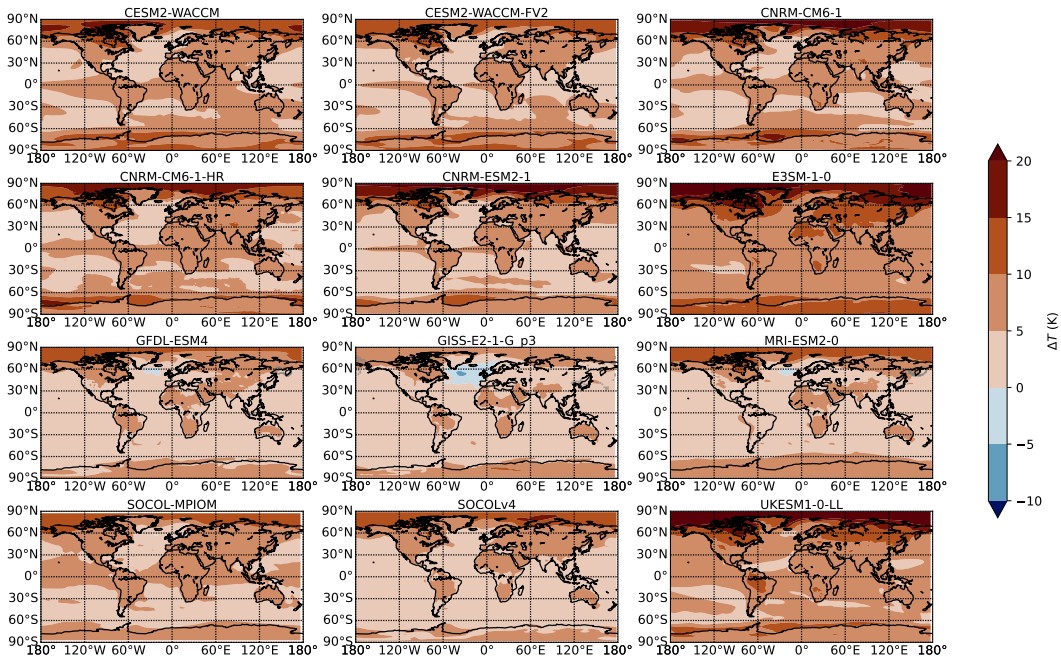

**Figure B3.** Annual-mean surface air temperature response to $4\times CO_2$. Regions that are not stippled are statistically significant.

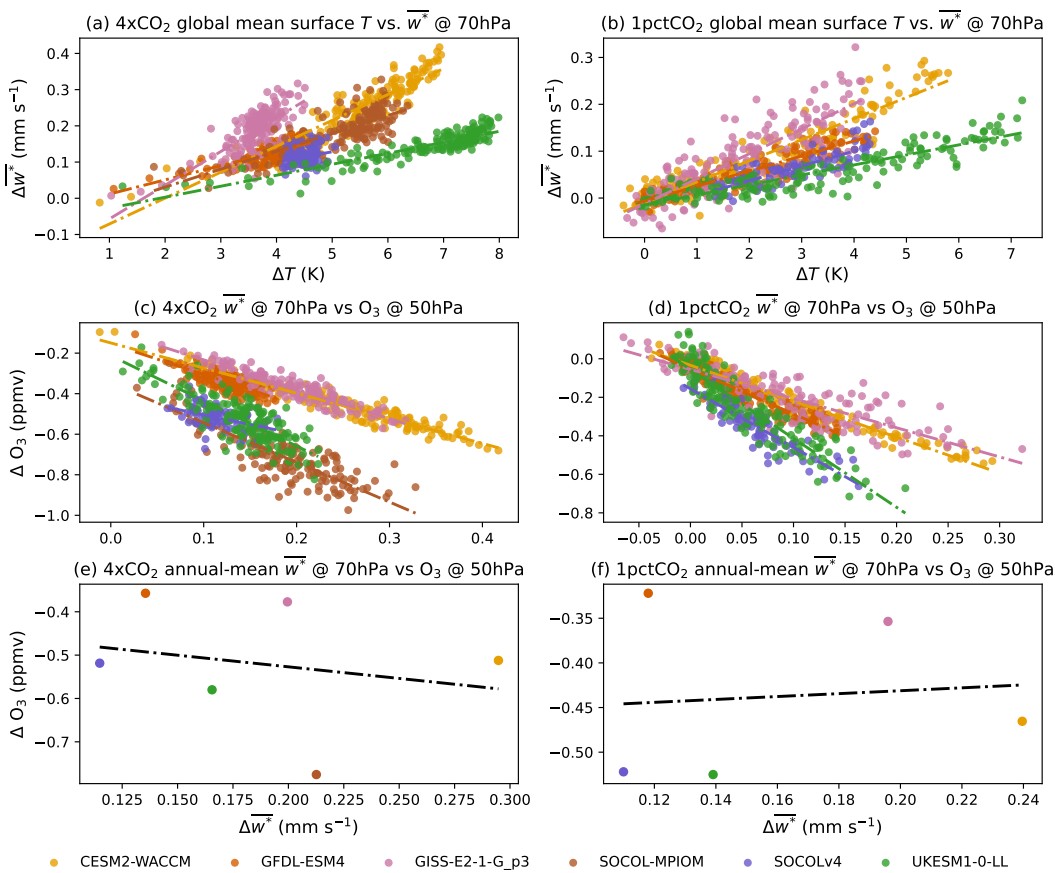

**Figure B4.** (a)-(b): Annual-mean tropical (15°S-15°N) residual upwelling ($\overline{w^*}$) response to global mean surface temperature for $4\times CO_2$ and $1pctCO_2$. (c)-(d):annual-mean tropical (15°S-15°N) ozone response at 50hPa to upwelling ($w$) change at 70hPa for $4\times CO_2$ and $1pctCO_2$. (e)-(f): same as (c)-(d), but averaged over the last 100 years for $4\times CO_2$ and for all the 150 years for $1pctCO_2$ experiment. Fitting lines retrieved from linear regression are plotted with the corresponding color of each model.

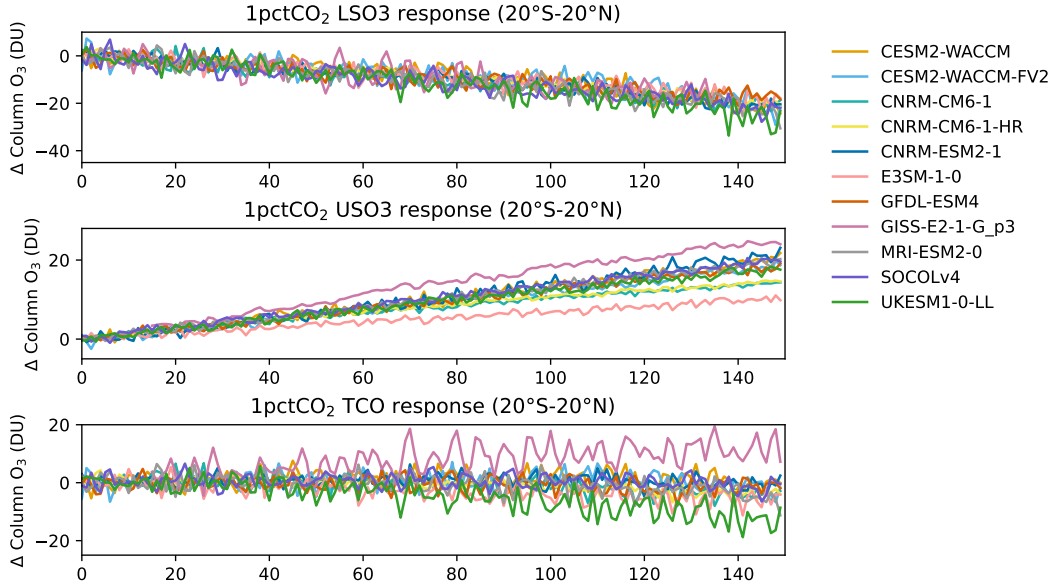

**Figure B5.** 150-year-long annual-mean tropical (20°S-20°N) LSO3, USO3 and TCO response to 1pctCO$_2$ with time.

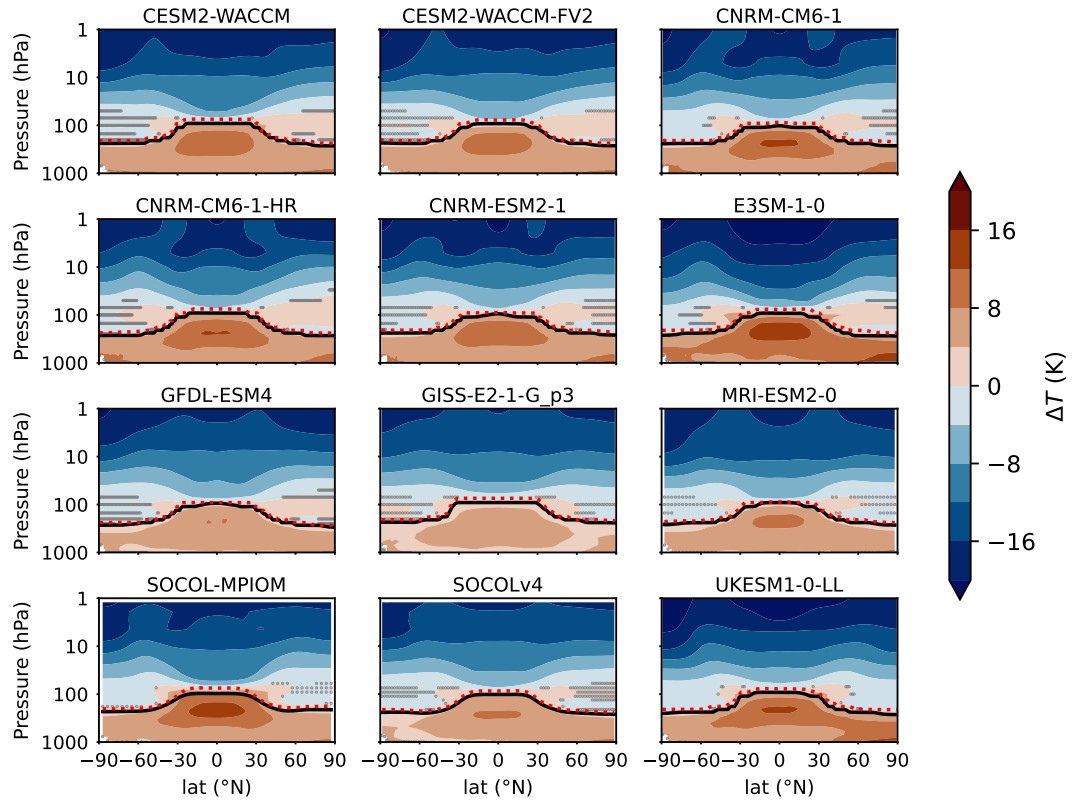

**Figure B6.** Annual-mean air temperature response to 4×CO$_2$ of each chem model.

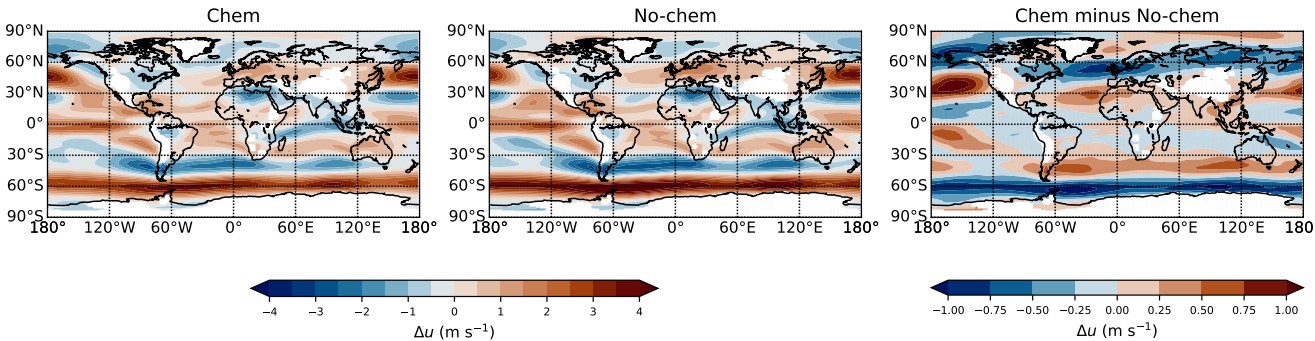

**Figure B7.** Multi-model average of annual-mean zonal wind response to 4×CO$_2$ at 850hPa in DJF for chem and no-chem models, along with the difference between the two categories of models.

**Table B1.** Residual tropical (20°S-20°N) upwelling ($\overline{w^*}$) change due to 4×$CO_2$ at 70 hPa (as provided on ESGF via the DynVarMip initiative) and average SSW frequency (identified based on Charlton and Polvani (2007) and calculated as number of SSW events per year) in piControl and 4×$CO_2$ in Chem vs. No-chem pairs.

| Pair (Chem/No-chem) | SSW (Chem/No-chem) | | $\overline{w^*}$ (Chem/No-chem) (mm s$^{-1}$) |
|---|---|---|---|
| | piControl | 4×$CO_2$ | 4×$CO_2$ - piControl |
| **CESM2-WACCM/CESM2** | 0.43/0.23 | 0.82/0.37 | 0.23/0.49 |
| **CESM2-WACCM-FV2/CESM2-FV2** | 0.52/0.4 | 0.71/0.32 | -/- |
| **GFDL-ESM4/GFDL-CM4** | 0.52/0.29 | 0.94/0.34 | 0.11/0.19 |
| **UKESM1-0-LL/HadGEM3-GC31-LL** | 0.59/0.83 | 0.42/0.29 | 0.13/0.16 |
| **SOCOLv4/MPI-ESM1-2-LR** | 0.72/0.95 | 1.02/0.65 | 0.10/- |

*Author contributions.* JW processed, analyzed the CMIP6 model datasets, run the SOCOLv4 model experiments, analyzed the SOCOLv4 data and drafted the manuscript. GC supervised JW in this project, helped with data analysis, and also discussion and edition of the manuscript. TS prepared the SOCOLv4 model experiments, discussed and edited the manuscript. BA conducted the SSW analysis. MD, BH, JK, PN, CO, SV and BA gave suggestions on the data and results analysis. All authors contributed to the preparation of the manuscript.

*Competing interests.* At least one of the (co-)authors is a member of the editorial board of the Atmospheric Chemistry and Physics journal. Besides this, we have no other competing interests to declare.

*Acknowledgements.* We acknowledge the World Climate Research Programme (WCRP) for coordinating and promoting CMIP6 through its Working Group on Coupled Modelling. We thank the climate modelling groups, who produced and made available their model output, the Earth System Grid Federation (ESGF) for archiving the data and providing access, and the multiple funding agencies who support CMIP6
and ESGF. We thank DynVarMIP for providing the TEM analysis metrics. We also thank the GFDL model development team, who led the development of GFDL-CM4 and GFDL-ESM4, as well as the numerous scientists and technical staff at GFDL who contributed to the development of these models and conducted the CMIP6 simulations. SOCOL simulations have been performed at the ETH cluster EULER and Swiss National Supercomputing Centre (CSCS) under projects s1191 and s1144. We thank Urs Beyerle (IAC ETH) for his help with acquiring and postprocessing the CMIP6 data. Jingyu Wang acknowledges support from the University of Arizona startup funds (PI: S. Ranjan).
Support for Gabriel Chiodo was provided by the Swiss National Science Foundation within the Ambizione grant no. PZ00P2_180043, the European Research Council within the ERC StG project no. 101078127, and the Spanish Ministry of Science and Innovation via the Ramon y Cajal grant no. RYC2021-033422-I. Support for Sandro Vattioni was provided by the ETH Research grant no. ETH-1719-2 as well as by the Harvard Geoengineering Research Program. Sandro Vattioni received also funding from the Simons Foundation (grant no. SFI-MPS-SRM-00005217). Timofei Sukhodolov acknowledges the support from the Swiss National Science Foundation (grant no. 200020E_219166)
and the Karbacher Fonds, Graubünden, Switzerland. Timofei Sukhodolov and Gabriel Chiodo also acknowledge the support from the Simons foundation (SFI-MPS-SRM-00005208). Birgit Hassler acknowledges the support from the European Union's Horizon 2020 research and innovation programme under Grant Agreement No. 101003536 (ESM2025—Earth System Models for the Future). Blanca Ayarzagüena acknowledges support from the project Stratospheric Ozone recovery in the Northern Hemisphere under climate change (RecO3very): PID2021-124772OB-I00 from the Spanish Ministry of Science and Innovation.

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
