# Peer review of "Exploring Ozone-climate Interactions in Idealized CMIP6 DECK Experiments"

_EGUsphere, 2025_

## Author Comment (AC3)

**Other changes: add GISS-E2-2-G into chem/no-chem comparison**

We added the chem (p3) and no-chem (p1) configuration of GISS-E2-2-G, which is the high-top version of GISS-E2-1-G as another chem/no-chem pair since it can better simulate BDC.

Here are the updated Figure 9-12 and Figure B7:

[Figure]

Figure 9. Comparison of the 100-year long annual-mean air temperature response to 4xCO2 between seven pairs of chem and no-chem CMIP6 models. The left column shows the response from chem models, the middle column shows the response from no-chem models and the right column shows the difference between chem and no-chem models.

[Figure]

Figure 10. Similar to Figure 9, but average of 100-year long annual mean air temperature response to 4xCO2 of six pairs of chem and no-chem models.

[Figure]

Figure 11. Comparison of the 100-year long seasonal mean zonal wind response to a 4xCO2 change in DJF between seven pairs of chem and no-chem models. Black contour lines depict the zonal wind climatology from the piControl experiment.

[Figure]

Figure 12. Similar to Figure 11, but a model average.

[Figure]

Figure B7. Multi-model average of annual-mean zonal wind response to 4xCO2 at 850hPa in DJF for chem and no-chem models, along with the difference between the two categories of models.

We also add discussion of GISS-E2-2-G as follows:
 In Methods section:

"We also examine results from the high vertical resolution version of the GISS CMIP6 climate model submission, GISS-E2-2 (Rind et al. (2020), Orbe et al. (2020)). Though identical in horizontal resolution to E2-1, E2-2 has more than twice the number of vertical levels (102) and a higher model top (0.002 hPa). This, in combination a non-orographic gravity wave drag scheme that is directly tethered to parameterized convection, produces in E2-2 more credible middle atmosphere dynamical and transport circulations, compared to observations (Orbe et al. (2020))."

In Results section, we add the discussion of Figure 11 as follows:

"The one exception is the E2-2-G model, in which interactive composition produces a more dramatic weakening of the Atlantic Meridional Overturning Circulation; this, on longer timescales,

accelerates the NH jet, obfuscating the initial weakening of the NH polar vortex (Orbe et al. (2024)). On shorter timescales, however, the initial response of the polar vortex in the interactive chemistry simulation is consistent with the response observed in the other models."

---

## Author Response (AR1)

**Responses to Reviewers**

We would like to thank the reviewers for their thoughtful views and valuable comments. Below is our response to each of the comment. The point-to-point responses are below with the reviewers' comments in BLACK, our responses in BLUE, and change in the manuscript in GREEN.

**Response to RC 1**

**General comments:**

1. The list of possible CMIP6 pairings of chemistry and no-chemistry models is incomplete: EC-Earth3 / EC-Earth3-AerChem could be added. My anticipation is that it would be worth adding this pair to the analysis.

**Response:** Thanks for the suggestion.
As described in van Nojie et al. (2021) (https://gmd.copernicus.org/articles/14/5637/2021/), the ozone field in EC-Earth3-AerChem is constrained using the CMIP6 forcing dataset from Checa-Garcia et al. (2018) (https://agupubs.onlinelibrary.wiley.com/doi/10.1002/2017GL076770). Specifically, the mixing ratios of ozone in the stratosphere are nudged towards zonal mean fields calculated from the three-dimensional input data sets provided by CMIP6 (Checa-Garcia et al., 2018). Therefore, by comparing the differences in the 4xCO2 response between this pair of models, we cannot infer the effects of the stratospheric ozone response. Thus, we decided not to add this pair to our analysis.

2. Furthermore, the authors state that there are differences other than the treatment of chemistry between these pairs. That is true for half the pairs but not the other. Perhaps something more profound can be said about how these other differences (resolution of middle / upper atmosphere, height of the model top, and tuning of the non-orographic gravity wave drag scheme, that characterize the CESM2 and GFDL pairs) affect model behaviour. To my understanding there are no substantial differences in anything other than chemistry between the HadGEM3/UKESM1, SOCOL4/MPIESM, and GISS pairs.

**Response:** Thanks for pointing it out.
Indeed, for the other three pairs (CESM2/CESM2-WACCM, CESM2-FV2/CESM2-WACCM-FV2, GFDL-CM4/GFDL-ESM4), they have other differences other than chemistry such as model top height. However, they share similar patterns in terms of the comparison between chem and no-chem with those pairs without major differences. This indicates that different chemistry scheme contributes the most to the chem/no-chem difference and other model differences play a minor role.

3. It is clear to me that most of the large role of climate-ozone interactions is due to the fact that in no-chemistry models the prescribed ozone field is not changing with the changing state of the atmosphere in the experiments considered here, unlike e.g. in "historical" simulations where ozone is amongst the external-forcing fields varying with time. Maybe this can be discussed, and whether the results of this study could motivate changes to the experiment definitions of 4xCO2 and 1pctCO2, where for no-chemistry models ozone could be made to change consistently with the evolving CO2 forcing, much like in "historical" simulations in future iterations of CMIP.

**Response:** Thanks for the suggestion.
Indeed, in CMIP 7, for abrupt-4xCO2 and 1pctCO2 experiments, it would worth trying to change the ozone forcing to the ozone field simulated in chem models for the corresponding experiment.

We added the following sentence to the conclusion section (P27 lines 428-429):
"Therefore, it might worth using the ozone field simulated in chem models as the forcing for no-chem models in future model intercomparison projects such as CMIP7."

**Minor comments:**
1. Table 2: As noted, the EC-Earth3 /EC-Earth3-AerChem pair can be added here.

**Response:** We decided not to add this pair due to the reason listed in the response to general comment #1.

2. Figure 1: Similar patterns of change were found by Morgenstern et al., ACP, 2018 (their figure 10), using CCMI1 models. They also documented similar inter-model differences to those seen here. However the mechanism discussed in the text (NOx production changes under climate change) may not have been represented in the older CCMI models, hence the pronounced increases in tropical-tropospheric ozone were not simulated. Perhaps this is worth a mention.

**Response:** Thanks for the suggestion.
Compared with Morgenstern et al., ACP, 2018, we think the tropospheric ozone increase doesn't seem more prominent in our analysis, but we added the follow sentence to the manuscript to discuss the potential role of NOx (P9 lines 173-174):

"A similar pattern was simulated in some of the CCMI1 models (Morgenstern et al., 2018), even though not all those models fully represent NOx production changes under climate change."

3. Figure 2: I find this figure hard to parse. A suggestion might be to calculate dO3/dT as a function of latitude and pressure for the various models and display that. Where these two

quantities do not highly correlate, this could be made NaN. Might that be a more intuitive way of displaying this information?

**Response:** Thanks for pointing it out.
dO3/dT as a function of latitude and pressure shows a similar pattern as depicted in the current version of Figure 1. However, we prefer to keep the current way of displaying the actual data since, if displayed along with Figure 1, one can infer the sensitivity and the actual change in the variable of interest. Moreover, it enables direct comparison between models by comparing the slopes. It also clearly shows the differences between different latitude bands and also different layers of the stratosphere, which helps readers understand the different dominant drivers of ozone responses. Therefore, we prefer to keep the current figure. We also updated the colormap to make it easier to read.

Here is the updated Figure 2:

[Figure]

Figure 2. 150-year-long annual-mean ozone response to temperature change in stratosphere at different pressure levels and latitude bands based on the 1pctCO2 experiment.

4. Figure 3: Indeed the relatively weak dependence of ozone on temperature is because of the low abundance of halogens in a PI world. There is no way the dots can be visually attributed to a particular model (not in my print-out, at least). Perhaps again a different way of displaying this can be considered?

**Response:** Thanks for the suggestion.
We updated the colormap to make it easier to differentiate between models. We further denote the value of R2 for each of the model in the legend to help readers understand the plot. The fitted lines show the correlation between ozone and zonal wind response, R2 indicates how strong this correlation is supported by the data, which are the main information we want to convey through this plot.

Here is the updated Figure 3:

[Figure]

Figure 3. 150-year-long seasonal-mean ozone response in 60-90N to zonal wind (u) change in 50-70N at 70hPa in MAM and DJF for 1pctCO2. Fitting lines retrieved from linear regression are plotted as dash-dotted lines with the corresponding color for each model. The thick black line is fitted using data from all models with the corresponding $R^2$ denoted in the upper right corner of the plot. $R^2$ values for each model are denoted in the legend for DJF and MAM respectively.

5. Figure 5: This figure is also similar to Morgenstern et al., ACP, 2018, their figure 11, showing essentially the same: Unambiguous increases in TCO in the northern extratropics, modeldependent signs of the tropical TCO trends due to cancellations, and a large spread of the ozone change over Antarctica.

**Response:** Yes, Figure 5 in our manuscript is similar to Figure 11 in Morgenstern et al., ACP, 2018. We added the citation to this paper in the following sentence in the manuscript (P15 line 258-260):

"These results are consistent with the analysis of the data from four CMIP5 models (Chiodo et al., 2018), including also the response in the NH being larger than that in the SH due to a stronger BDC (Butchart et al., 2014). They are also largely consistent with a previous study using CCMI-1 data on the sensitivity of ozone to GHGs (Morgenstern et al., 2018)."

**Response to RC 2**

**General comments:**
The authors pointed out that Arctic ozone increase when the Arctic stratospheric vortex weakens. The study uses multiple CMIP6 models to analyze the relationship between ozone and the polar vortex; however, the models differ significantly in their simulations of polar vortex strength, suggesting that there may be uncertainty in key processes within the models. The authors suggest that this relationship is stronger in winter but weaker in spring on the interannual timescale. But from the perspective of seasonal variation, a weakening of the barrier in early spring may lead to enhanced transport, so why is the response weaker in this period? Actually, the breakup of polar vortex associated with final warming during early spring is also closely related to the transport barrier effect. The authors shall investigate the connection of breakup time of polar vortex in early spring to ozone changes, instead of using March-April-May mean, which may mask this relationship. In a short, I think the sentence of 'the transport barrier role of the polar vortex is generally weaker in spring than in winter' is not appropriate. In addition, the Antarctic polar vortex is stronger and more stable than the Arctic polar vortex, why is there no discussion of how changes in the Antarctic polar vortex respond to ozone feedbacks?

**Response:** Thanks for the comments and the suggestions.
Indeed, averaging over a long time span might blur the relationship, and thus it is not appropriate to infer the response of Arctic stratospheric vortex from this analysis. In order to study the breakup time of polar vortex, one would need to align the data relative to the final warming date of each individual model, which is out of the scope of this paper Therefore, we revised our discussion of Figure 3 to emphasize this caveat as follows (P12 lines 225-228):

"The breakup of the polar vortex may lead to enhanced transport of ozone to polar region, but averaging over MAM may mask this relationship. Investigation of the breakup time of polar vortex and how it changes under climate change would need to be considered for each models, which is out of scope, but which merits further investigation."

We also added a similar plot as Figure 3 for Antarctic vortex and corresponding discussion in the appendix as following:

[Figure]

Figure B2. 150-year-long seasonal-mean ozone response in 60-90S to zonal wind (u) change in 50-70S at 70hPa in SON and JJA for 1pctCO2. Fitting lines retrieved from linear regression are plotted as dash-dotted lines with the corresponding color for each model. The thick black line is fitted using data from all models with the corresponding R2 denoted in the upper right corner of the plot. R2 values for each model are denoted in the legend for SON and JJA respectively.

**Minor comments:**

1. Line2: '…, and lead to' -> '…, leading to'

**Response:** Revised accordingly.

2. Line6: 'This work employs the latest data from Coupled Model Intercomparison Project Phase 6 (CMIP6), …' The comma after "CMIP6" is unnecessary.

**Response:** Revised accordingly.

3. Line10: 'We then explore the feedback exerted by ozone on climate'. This expression can be simplified as 'We then explore the ozone-climate feedback'

**Response:** Revised accordingly.

4. Lines11-12 'We find that the stratospheric temperature response is substantial, with a global negative radiative forcing by up to −0.2 W m−2.' The radiative forcing responses of the different models have large variations, and in the text analysis shows that the largest radiative forcing is −0.19 W m−2 and is derived from the UKESM1-0-LL that does not perform well in any of the other feedback processes (including ozone response to 4×CO2, ozone response to temperature change and SSW frequency change due to 4×CO2), and I think that a clear range of global mean net radiative forcing should be included in the abstract.

**Response:** Thanks for the suggestion.
We replaced "with a global negative radiative forcing by up to −0.2 W m−2" with "with a global negative radiative forcing ranging from -0.03 W m-2 to -0.19 W m-2".

5. Line152: 'against' -> 'with'

**Response:** Revised accordingly.

6. Line158: 'year 135 to 145' -> 'years 135 to 145'

**Response:** Revised accordingly.

7. Line172: It should be 200-240 nm in this reference.

**Response:** Revised accordingly.

8. Line 209: Figure 3 only reflects the correlation between ozone and zonal wind. How did you know that the polar vortex is weakening from Figure 3? Is it through the average zonal wind of each model?

**Response:** Thanks for bringing it up!
We mainly look at the correlation between the ozone and zonal wind response indicated by the negative trend instead of the absolute change, which is shown later in Figure 11. From this correlation, we propose that when the polar vortex is weakened (delta_u <0), the weakened barrier leads to more mixing of polar air with ozone-rich air, thus an increase in ozone abundance for most models.

We have revised the corresponding sentences as following to make it clearer (P12 lines 218-220):
"Figure 3 shows that in winter, for most models, the weakening of the NH polar vortex reflected by the weakened zonal winds in 50-70N correlates with an increase of ozone in the Arctic (small but significant negative slope)."

9. Line223: in most locations -> in most regions

**Response:** Revised accordingly.

10. Line243: Decoupling -> Decomposing

**Response:** Revised accordingly.

11. Line259: 'during the last 80 years' perhaps it could be changed to 'over the subsequent 80 years'

**Response:** Revised accordingly.

12. Line354: Do you mean stratospheric ozone depletion or stratospheric ozone recovery? There are different behaviors of the polar vortex and jet stream under these two scenarios. Please clarify it.

**Response:** Since we are comparing chem/no-chem, the change would stem from the changes in ozone under 4xCO2, which is similar (but not necessarily the same as) future stratospheric ozone recovery.

13. Line410: Expanded AMOC as "Atlantic Meridional Overturning Circulation" when first introduced in the sentence, then used the abbreviations consistently.

**Response:** Revised accordingly.

14. Although the authors mention statistical significance tests (e.g., t-tests), there is limited information on the exact methods used. It would be useful to provide more details about the statistical.

**Response:** Thanks for bringing it up.
We add the following sentences in the Results section to explain how we did the t-test (P8 lines 166-168):
"We assume the timeseries of ozone concentration under piControl and 4xCO2 are independent samples with the same variance, then we compute the t statistic to see if the two samples have same mean value. This also applies to other variables we analyze hereafter."

15. The manuscript provides a detailed assessment of the long-term (150-year) ozone response to increased CO2. Meantime, the authors mention that ozone changes in the early stages of CO2 increase are characterized by rapid adjustment. Does this fast-adjusting response exhibit

nonlinearities or threshold points? Could this threshold point depends on whether the chem or no-chem model? Is there some consistency of threshold in the chem/no-chem models?

**Response:** Thanks for the question.
We do not find evidence of any nonlinearities or threshold points from the evolution of ozone in the 1pctCO2 experiment. And therefore, we don't expect any nonlinearities in the chem/no-chem models. There is also no clear evidence of non-monotonic behavior in the stratospheric circulation under increasing $CO_2$ forcing, at least for GISS (Menzel et al., 2023 https://doi.org/10.1175/JCLI-D-22-0851.1).
However, the only aspect that introduces some non-linearity might be the AMOC, which collapses at different times across models and configurations, and in one case (GISS), the difference in the behavior may be related to ozone feedbacks (Orbe et al. 2024 https://doi.org/10.1175/JCLI-D-23-0119.1), but this has not been shown yet for other models.

16. The different colors in Fig.2 are hardly to see. Please redraw it.

**Response:** Thanks for pointing it out.
We have updated Figure 2 as following:

[Figure]

Figure 2. 150-year-long annual-mean ozone response to temperature change in stratosphere at different pressure levels and latitude bands based on the 1pctCO2 experiment.

---

## Author Response (AR2)

**Response to Editor**

We would like to thank the editor for their thoughtful views and valuable comments. Below is our response to each of the comment. The point-to-point responses are below with the editor's comments in **BLACK**, our responses in **BLUE**, and change in the manuscript in **GREEN**.

1. L. 199: "differences in the photochemistry schemes" -> also radiative heating schemes.

**Response:** Thanks for the suggestion.

We modified the original sentence to consider radiation scheme, which includes both radiative heating and cooling scheme.

We replace the original sentence with the sentence below (P10 lines 198-200):

"…and differences in the photochemistry and radiation schemes of the models explain the different sensitivities of ozone to temperature change across models…"

2. L. 206-210: stronger tropical upwelling also means less ozone poor tropospheric air gets to the lower stratosphere.

**Response:** Thanks for pointing it out!

We have modified the sentence as follows (P10 lines 207-208):

"The stronger upwelling results in enhanced transport of ozone out of the tropical pipe and ozone-poor air from the troposphere to the LS, leading (locally) to a decrease."

3. L. 218-219 – this sentence in the current form still implies that models project a weakening of NH polar vortex under climate change, which is not true (see e.g. Karpechko et al., 2022). Please rephrase to "a weaker NH polar vortex correlates with an increase of ozone in the Arctic", to similar.

**Response:** Thanks for the suggestion!

We have modified the sentence as follows (P12 line 219):

"…for most models, a weaker NH polar vortex correlates with an increase of ozone in the Arctic."

4. L. 221 "the heating from more ozone" – but we're talking about DJF and there is no sunlight in the Arctic in DJF, and so any radiative impacts of Arctic ozone (if at all) would have to come from LW cooling.

**Response:** Thanks for the comment! It is a good point.

The reason that we focus on LW heating is due to the correlation of the weakening of zonal wind with the increase of ozone response shown in Figure 3. We also elaborate more to discuss about the dominant role that LW cooling can play.

We modified the sentence and added a new sentence as follows (P12 lines 223-224):

"Also, the heating from more ozone could partly contribute to polar warming and weakening of the polar vortex throughout the extended winter season, and there is more ozone in the surf zone available to be transported into the polar region. Conversely, during the deep winter months, LW cooling effect from ozone anomalies can be dominant at the polar-most latitudes."

5. Please make sure that from Section 3.1.2 onwards, the numbering of figures in the Appendix (as indicated by Fig. Bx) is corrected and incremented by 1 throughout the rest of the manuscript (as new Fig. B2 was added but the subsequent figures are not correctly referenced now).
**Response:** Thanks for catching that! We have revised accordingly.

6. L. 243: "Climatological differences" -> "Climatological responses"
**Response:** Thanks for the suggestion! We have revised accordingly.

7. L. 244: add "under higher GHGs" (or similar) at the end of the sentence.
**Response:** Thanks for the suggestion! We have revised accordingly.

8. L. 245: "climatological spread of the two variables" -> "differences in the responses in the two variables" (or else, currently unclear in the present form)
**Response:** Thanks for the suggestion!

We have revised as follows (P13 lines 246-248):

"…we don't find a consistent relationship in the inter-model differences in the responses in the two variables…"

9. Section 3.2. Please define TCO, TRO3, USO3 and LSO3.
**Responses:** Thanks for the suggestion. We have revised accordingly.

10. L 253 "a high bias" – you mean it's an outlier? Because we can't see it's a bias as we don't know what the 'correct' value is in this case, we can only say the other models show either near zero or negative responses.
**Response:** Thanks for pointing it out!

We have modified the sentence as follows (P15 lines 253-255):

"In the tropics, the multi-model uncertainty is smaller, though UKESM1-0-LL and SOCOL-MPIOM show a more negative TCO response and GISS-E2-1-G_p3 is an outlier in the opposite direction."

11. L. 271. It might be worth stating which experiment has higher CO2 around year 140, or if they both have similar CO2 levels at that time.
**Response:** Thanks for the suggestion.

The two experiments have similar CO2 level at year 140, and we have clarified that in the text as follows (P16 lines 273-275):

"…we can see the lag of the response since the tropical TCO response around year 140, when the CO2 level in 1pctCO2 reaches that in 4CO2, is smaller than the equilibrated value from 4CO2."

12. L. 285 – 286. It's not obvious to me that the 'full' response in models in the NH midlatitude lower stratosphere is necessarily a warming. To me it looks like the temperature response in the mid-latitude LS is as uncertain as in the high latitude LS.

**Response:** Thanks for pointing it out!

We have edited the sentence to make it more precise (P17-18 Lines 289-291):

"In the LS, we find a slight warming in the NH middle and high-latitudes, contributing to the total temperature response in this region, competing with a similar level of cooling from CO2, which results in differences in the sign of the total temperature response across models."

13. L. 300 "LW flux changes largely corresponds to the temperature adjustments in the LS and is consistent with the ozone changes" – While changes in T (caused by O3) definitely lead to changes in LW, changes in O3 themselves also lead to changes in LW (so not just by changing T but also by more/less LW absorption and re-radiation).

**Response:** Thanks for pointing it out!

We have added a sentence to discuss the effect of ozone response on LW flux (P19 lines 305-306):

"The ozone changes determine the temperature adjustment and modulate the LW flux by increasing the local LW absorption and emission."

14. L. 309-310. Might be worth pointing out also that it's the tropical response that dominates the global mean response, and that is always negative (unlike in the mid/high latitudes)

**Response:** Thanks for the suggestion!

We have modified the sentence as follows (P19 lines 314-315):

"The net flux change is the sum of that of LW and SW, and is negative in the tropics and positive in the extratropics, with the LW generally dominating over the SW forcing. Due to the large area of the tropics, the global mean net RF is negative in all models…"

15. L.360-361: "The strengthening of zonal wind in the tropical US is due to the expansion of the weakened polar vortex as discussed in Section 3.1.1." – I'm confused by both what strengthening of the zonal wind in tropical US and also what expansion of the weakened polar vortex. Please either clarify or delete this sentence.

**Response:** Thanks for pointing it out!

We have clarified that in the following sentence (P23 lines 366-367):

"The positive zonal wind anomalies in the tropical US could be related to the expansion of the polar vortex area due to its weakening, as discussed in Section 3.1.1."

16. L. 405-406: "increase of Arctic ozone is partly due to weaker westerlies, and thus more in-mixing of ozone rich air into the polar region". I disagree. First, as I said earlier, models don't show any consistent sign of projected changes in the strength of Arctic polar vortex under climate change (Karpechko et al., 2022), with some

showing jet strengthening, some showing jet weakening, and some not showing any change at all (this is seen also in your Fig. 11). Second, any changes in Arctic polar vortex would really only matter in winter (and maybe late autumn and early spring). What determines the year-round polar ozone increase under higher CO2 is strengthening of the BDC, and the resulting enhanced transport of ozone rich air from the production region (tropics) to mid and higher latitudes, plus the overall increase in O3 in mid/upper stratosphere from GHG-induced cooling. Please change.

**Response:** Thanks for the suggestion.

We have modified the sentence as follows (P26 lines 411-414):

"In the lower stratosphere, there is a decrease of ozone dominated by stronger upwelling, while in the middle and high latitudes, the ozone anomalies are positive, because of the combination of increased transport by BDC and overall higher ozone abundance due to cooling from CO2. Additionally, the weakening of westerlies also leads to more in-mixing of ozone-rich air into the polar region during winter."

17. L. 407. Please say this is true for the multi-model mean, and that individual models can show both positive, negative or near zero changes.

**Response:** Thanks for the suggestion.

We have modified the sentence as follows (P26 lines 415-416):

"In the multi-model mean sense, the total column ozone response is negligible in the tropics because of the cancellation between decreases in the lower stratosphere and increases in the upper stratosphere."

18. L. 409. "which is related to the different response of zonal wind in the models". As above, the main driver of these differences is different response of BDC in these models, not zonal wind (although zonal wind response would obviously have some influence too, but it will be smaller, especially in the Artcic, than BDC)

**Response:** Thanks for pointing it out!

We have modified the sentence as follows (P26 lines 416-418):

"Total column ozone increases at high latitudes, but with a larger inter-model discrepancy compared to the tropics, which is related to the differences in the response of residual ozone transport in the models and, to some extent, uncertainty in the polar vortex response."

19. L. 418-420. Again, I disagree with this sentence as currently written, as it implies that the models project weakening of the polar vortex under increased CO2, which is not true. What the results show is that the inclusion of ozone feedback leads to a more easterly response of the stratospheric polar vortex in both hemispheres. Please correct.

**Response:** Thanks for pointing it out!

We have modified the sentence as follows (P27 lines 426-429):

"These temperature changes cause the weakening of the stratospheric polar vortex during boreal winter, with this signal extending to the troposphere. Compared to the model uncertainty in the total response of polar vortex to increased $CO_2$,the models are more consistent in terms of the sign of the response."

20. L. 428 "it might worth" -> "it highlights the need for". It might worth is way too weak given the whole paper demonstrates why is it so important to include ozone.

**Response:** Thanks for the suggestion. We have revised accordingly.

21. L. 433. "and weaker westerlies in polar region". Please delete as this not true (as stated above).

**Response:** Thanks for the suggestion. We have revised accordingly.

22. L. 444: might be worth adding "direct" to "radiative effect of ozone response"

**Response:** Thanks for the suggestion. We have revised accordingly.

23. Figure B2 – swap panel order, JJA should come before SON

Response: Thanks for the suggestion. We have revised accordingly.